# Growth in fluctuating light buffers plants against photorespiratory perturbations

Thekla von Bismarck [1,2,3] ✉, Philipp Wendering [3,4], Leonardo Perez de Souza [3], Jeremy Ruß[3], Linnéa Strandberg [3], Elmien Heyneke[3], Berkley J. Walker [5,6], Mark A. Schöttler [3], Alisdair R. Fernie [3], Zoran Nikoloski [3,4] & Ute Armbruster [1,2,3] ✉

Photorespiration (PR) is the pathway that detoxifies the product of the oxygenation reaction of Rubisco. It has been hypothesized that in dynamic light environments, PR provides a photoprotective function. To test this hypothesis, we characterized plants with varying PR enzyme activities under fluctuating and non-fluctuating light conditions. Contrasting our expectations, growth of mutants with decreased PR enzyme levels was least affected in fluctuating light compared with wild type. Results for growth, photosynthesis and metabolites combined with thermodynamics-based flux analysis revealed two main causal factors for this unanticipated finding: reduced rates of photosynthesis in fluctuating light and complex re-routing of metabolic fluxes. Only in non-fluctuating light, mutants lacking the glutamate:glyoxylate aminotransferase 1 re-routed glycolate processing to the chloroplast, resulting in photooxidative damage through $H_2O_2$ production. Our results reveal that dynamic light environments buffer plant growth and metabolism against photorespiratory perturbations.

The fixation of carbon dioxide ($CO_2$) into biomass is one of the most essential processes for life on Earth. The responsible enzyme ribulose biphosphate carboxylase/ oxygenase (Rubisco) cannot entirely discriminate between $CO_2$ and molecular oxygen ($O_2$) and thus, as a side reaction, fixes $O_2$ onto the acceptor molecule ribulose-1,5-bisphosphate. While the carboxylation reaction of Rubisco results in the production of two molecules of 3-phosphoglycerate, the oxygenation reaction produces only one of these molecules together with one molecule of 2-phosphoglycolate (2PG). 2PG cannot be directly fed back into the Calvin-Benson-Bassham (CBB) cycle and thus its accumulation drains molecules from the CBB cycle. Additionally, 2PG inhibits several chloroplast enzymes, together leading to the arrest of photosynthetic reactions and lethality of the plant when 2PG accumulates to high levels[1–3].

The photorespiratory pathway (PR) metabolizes 2PG and regenerates intermediates of the CBB cycle. Its reactions span four different compartments, eight core enzymes and multiple transporters that shuttle the intermediates between different compartments (Fig. 1a; reviewed in ref. 4). The PR reactions result in the release of $CO_2$ and ammonia and come at a high energy cost[5,6]. Thus, current plant engineering efforts aim to replace PR through bypasses that retain carbon and/or nitrogen or have lower energy requirements (e.g., refs. 7–14). Metabolic reactions of PR do not only contribute to the recycling of 2PG, but also feed into other metabolic processes, including $C_1$ metabolism[15], nitrogen fixation[16,17], sulfur metabolism[16,17], subcellular re-distribution of energy equivalents[18], tricarboxylic acid (TCA) cycle and respiration[19,20].

[1]Molecular Photosynthesis, Heinrich-Heine-University Düsseldorf, Universitätsstraße 1, 40225 Düsseldorf, Germany. [2]CEPLAS - Cluster of Excellence on Plant Sciences, Heinrich Heine University Düsseldorf, Düsseldorf, Germany. [3]Max Planck Institute of Molecular Plant Physiology, Am Mühlenberg 1, 14476 Potsdam, Germany. [4]Bioinformatics Department, Institute of Biochemistry and Biology, University of Potsdam, Karl-Liebknecht-Str. 24-25, 14476 Potsdam, Germany. [5]DOE-Plant Research Laboratory, Michigan State University, 612 Wilson Rd, East Lansing, MI 48824, USA. [6]Department of Biochemistry and Molecular Biology, Michigan State University, 603 Wilson Rd Rm 212, East Lansing, MI 48823, USA. ✉e-mail: thekla@v-bismarck.de; ute.armbruster@hhu.de

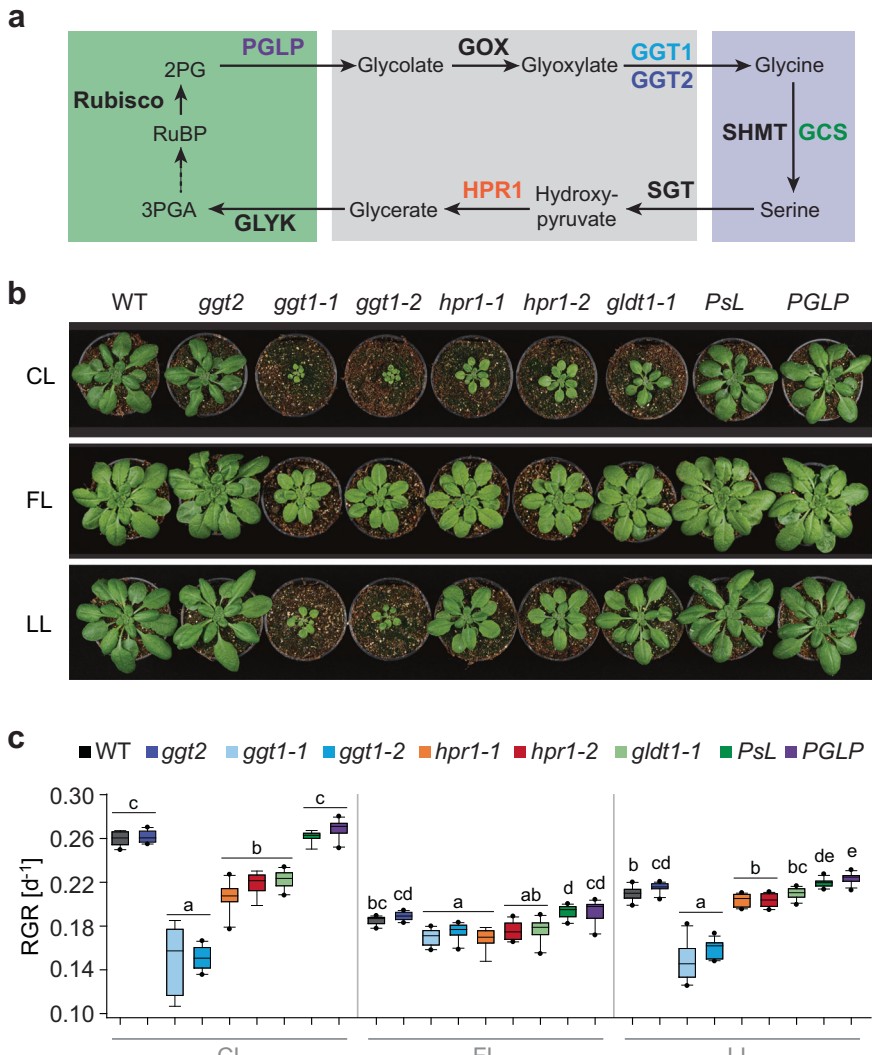

**Fig. 1 | Distinct effects of light conditions on different photorespiratory mutants. a** Scheme of the photorespiratory pathway spanning the chloroplast (green), peroxisome (grey) and mitochondrion (purple). Proteins (in bold) with altered levels in the specific plant lines are indicated in the same color as the corresponding plant lines in panel c. PGLP – 2-phosphoglycolate phosphatase; GOX – Glycolate oxidase; GGT1/2 – Glutamate:glyoxylate aminotransferase 1/2; SHMT – Serine hydroxymethyltransferase; GCS – Glycine cleavage system; SGT – Serine:glyoxylate aminotransferase; HPR1 – Hydroxypyruvate reductase 1; GLYK – D-glycerate 3-kinase. **b** Images of WT, *ggt2*, two mutant alleles of each *ggt1* and *hpr1*, *gldt1-1* (knockdown of the gene encoding for the T protein of the GCS, accumulates only 5% of WT protein) and two overexpression lines *PsL* (encoding the L protein of the GCS, ~250% of enzyme activity compared with WT) and *PGLP* (~150% of enzyme activity compared with WT) grown in control light (CL: 200 μmol photons m⁻² s⁻¹),

fluctuating light (FL: 1 min 700 μmol photons m⁻² s⁻¹, 4 min 70 μmol photons m⁻² s⁻¹) or low light (LL: 90 μmol photons m⁻² s⁻¹) for 29, 43 or 37 days after sowing, respectively. **c** Relative growth rates (RGR), i.e., the rate of accumulation of new dry mass per unit of existing dry mass per day, of all lines as described in panel b. Each box is based on n = 7 (CL: *ggt1-1*), n = 9 (CL: *hpr1-2*, *PsL*; FL: *hpr1-1*), n = 10 (CL: WT, *ggt2*, *ggt1-2*, *hpr1-1*, *gldt1-1*, *PGLP*; FL: WT, *ggt2*, *ggt1-1*, *ggt1-2*, *hpr1-2*, *gldt1-1*, *PsL*, *PGLP*; LL: *ggt1-1*), n = 11 (LL: *hpr1-1*) and n = 12 (LL: WT, *ggt2*, *ggt1-2*, *hpr1-2*, *gldt1-1*, *PsL*, *PGLP*) replicates. The lower and upper boundaries of the box indicate the 25th and 75th percentile, the median is shown by the middle line. Whiskers indicate maximum and minimum and black circles represent outliers that fell outside the 10th and 90th percentiles. Different lowercase letters above boxes indicate significant differences between groups within one condition as determined by the Games–Howell multiple comparison test.

Under natural sunlight conditions, plants are often exposed to strong and rapid fluctuations in light intensity[21]. These fluctuations induce regulatory mechanisms, which are important for plant fitness in the field[22]. The responses of some of these regulatory mechanisms are much slower than changes in light intensity and it was proposed that kinetic inefficiencies may limit crop yield[23–27]. Directly following a transition from low to high light, delays in the induction of stomatal conductance and the CBB cycle lead to a transient decrease in $CO_2$ supply to Rubisco[21,28] and overproduction of ATP and NADPH that cannot be used by the CBB cycle. It was hypothesized that, in fluctuating light, PR plays a role for counteracting the kinetic inefficiencies of these two processes, by: (i) providing increased rates of 2PG metabolization needed because of higher Rubisco oxygenation per

carboxylation rates[29] and (ii) simultaneously serving a sink for the transient spikes of metabolic energy produced by the light reactions, thereby avoiding photo-oxidative damage to the photosynthetic machinery[29–31]. For *Arabidopsis thaliana* (Arabidopsis) grown in fluctuating light, it was shown that they accumulated higher transcript and protein levels of some PR enzymes, suggesting that PR metabolism is upregulated[32,33].

A complete block in PR causes Arabidopsis plants to die when grown in air (with ambient $CO_2$ concentration)[34,35]. Arabidopsis mutants devoid of either the peroxisomal glutamate:glyoxylate aminotransferase 1 (GGT1) or the hydroxypyruvate reductase 1 (HPR1) survive in air, as there are redundant enzymatic reactions/ pathways that can at least partially compensate for their loss[36–40]. Mutants with a

knockdown in the gene encoding the T protein of the mitochondrial glycine cleavage system (GCS), *gldt1-1*, showed a comparable growth phenotype with *hpr1* and *ggt1* in air[41]. Thus, these mutants are considered "mild" PR mutants[42].

In the present study, we investigated mutants with varying levels of several PR enzymes to determine effects of altered PR flux on plant metabolism and growth, comparing non-fluctuating with fluctuating light conditions. By combining gas exchange measurements, metabolite analysis, and thermodynamic-based flux analysis, we reveal that PR metabolism is highly flexible and alternative pathways are differentially engaged in response to the light conditions. Intriguingly, this flexibility contributes to the nearly unimpaired growth of mutants with restrictions in PR under fluctuating light conditions.

## Results

### PR genes cluster as part of a larger co-expression network encompassing regulators of photosynthesis

We found photorespiratory genes, including that encoding the close *GGT1* homolog *GGT2*, to be co-regulated at the transcript level with regulators of photosynthesis in fluctuating light, i.e., the thylakoid K⁺ exchange antiporter 3, the serine/ threonine-protein kinase STN7 and the zeaxanthin epoxidase[43–45] (Supplementary Fig. 1a). This finding sparked our interest to investigate the functional relevance of PR in dynamic light conditions. Because of the co-expression-based link of *GGT2* with the regulation of photosynthesis, we performed initial analyses on GGT2 localization and expression. GGT2 has a predicted peroxisomal targeting peptide, and our data support a localization in small compartments within the cell, that may be peroxisomes (Supplementary Fig. 1c, e). Publicly available transcriptomics data suggest that *GGT2* has a minor function in the photosynthetic leaf tissue as compared to *GGT1*, because it is less expressed (Supplementary Fig. 2).

### Photorespiratory mutants grow comparably well in fluctuating light conditions

To investigate the role of PR in dynamic light, we grew Arabidopsis plants with different modifications of PR metabolism under one strongly fluctuating light and two non-fluctuating light regimes.

Besides wild type (WT), a newly characterized *ggt2* T-DNA insertion line (Supplementary Fig. 1d), the previously published "mild" PR mutants *ggt1*, *hpr1* and *gldt1-1*[36,39–41], and two lines with reported higher PR flux (overexpression of *PGLP* or the L protein of GCS, i.e., *PsL*[3,46]) were used (Fig. 1a). The growth light was provided at a 12 h photoperiod with 144 repetitions of 10-fold changes in light intensity (4 min 70 μmol photons $m^{-2}$ $s^{-1}$ and 1 min 700 μmol photons $m^{-2}$ $s^{-1}$) for the strongly fluctuating light (FL) condition, 200 μmol photons $m^{-2}$ $s^{-1}$ for the control light (CL) condition with the same average daily photon flux density as FL, and 90 μmol photons $m^{-2}$ $s^{-1}$ for a low light (LL) control condition intended to account for similar rates of $CO_2$ fixation as in FL.

Relative growth rates per day (RGR) were calculated from the dry weight of the plants (Fig. 1b, c, Supplementary Figs. 3, 4). Together, these analyses revealed that the PR mutants *ggt1*, *hpr1,* and *gldt1-1* grew significantly slower than WT under CL, while *ggt2* and the overexpressors grew WT-like. Notably, the growth retardation of *ggt1*, *hpr1* and *gldt1-1* compared with WT were much less severe when grown in FL (Fig. 1b, c; Supplementary Fig. 4). Both *ggt1* mutants also showed severely reduced growth under LL, while this was not the case for *hpr1* and *gldt1-1*. Additionally, *PGLP* overexpression increased growth rates consistently only under the LL condition.

These growth analyses showed that plants lacking the enzymes GGT1 or HPR1, or harboring decreased levels of GCS, display similarly mild growth defects under FL. Growth under non-fluctuating light conditions instead revealed that mutants fell in two categories, with *hpr1* and *gldt1-1* growing much better than *ggt1*, particularly under LL.

### Oxygen-suppressed $CO_2$ assimilation as a proxy for PR rates can explain the light condition-dependent *hpr1*, but not the *ggt1* growth phenotype

We then investigated whether the difference between growth phenotypes of the PR deficient mutants *ggt1* and *hpr1* under the different light environments were related to photosynthetic rates and thus 2PG production by the Rubisco side reaction. Therefore, $CO_2$ assimilation rates ($A$) were measured under photorespiratory and non-photorespiratory conditions (i.e., in air and in low oxygen with 21% and 2% $O_2$, respectively), supplying a light regime that was replicated from the respective growth light (Fig. 2a, Supplementary Fig. 5). The increase in $A$ under low oxygen served as a proxy for the rates of oxygen fixation in air (i.e., oxygen-suppressed $A$; $A_{sup}$).

In line with faster growth of WT in CL compared with LL, $A$ was highest in CL across genotypes (Fig. 2a). In air, both *ggt1* and *hpr1* displayed a significant reduction of $A$ compared with WT in CL (by ~18% and ~29%, respectively), while only *hpr1* showed a significant decrease also in FL (by ~22%; Fig. 2a). Generally, rates of $A_{sup}$ increased proportionally with $A$ (Pearson correlation coefficient: 0.58, $p < 0.0001$). Within growth conditions, there was no significant genotype-specific effect on $A_{sup}$ (Fig. 2b). Together, these data support that of all three conditions Rubisco activity and thus rates of 2PG production were highest in CL. WT and *ggt2* had similar $A$ in FL as in LL, which matched the comparable growth rates under both conditions (slightly higher in LL).

For WT and *ggt2*, mean internal $CO_2$ concentrations ($C_i$) were similar between LL and FL and significantly higher than in CL (Fig. 2c, Supplementary Fig. 6). Both *ggt1* alleles showed significantly lower $C_i$ in LL compared with FL, while $A$ was comparable (Supplementary Data 1). For *hpr1*, we found $A$ to be significantly higher in LL than in FL, while $C_i$ was similar (significantly higher in FL as compared to LL only in the *hpr1-2* allele; Supplementary Data 1). The similar $A_{sup}$ and $C_i$ values across genotypes for each condition suggested that relative rates of oxygenation per carboxylation reactions were comparable to WT in both *hpr1* and *ggt1* (Supplementary Data 1). All gas exchange parameters determined for *ggt2* were WT-like. The small increase in $C_i$ from 340 (CL) to 360 ppm (FL) did not efficiently repress photorespiration, as evidenced by similar $A_{sup}$ between both conditions and therefore cannot explain the better growth of the mutants under FL relative to CL.

The results of these gas exchange measurements support a model in which plant growth in FL is less affected in *ggt1* and *hpr1*, because residual PR flux in these mutants is sufficient to turnover most of the produced 2PG. In CL, however, rates of 2PG production exceed the capacity of residual PR in *ggt1* and *hpr1*, resulting in reduced CBB cycle activity and negative effects on plant growth. Additionally, increased non-enzymatic $CO_2$ release from hydroxypyruvate or glyoxylate or increased flux through the oxidative pentose phosphate pathway as suggested previously for *hpr1*[36,37,47,48] may further reduce net $CO_2$ assimilation. While improved growth of *hpr1* in LL is in line with this interpretation, it cannot explain the strong growth retardation of *ggt1* under this condition.

### Restricted PR metabolism negatively affects photosystem II efficiency after FL to CL shifts

We obtained spatial and time-resolved information on the maximum quantum efficiency of PSII ($F_v/F_m$) by using an Imaging-PAM during shifts from FL to CL or LL. FL was chosen as the pre-shift condition because the growth of the "mild" PR mutants was most comparable with WT. $F_v/F_m$ correlates with the intactness of PSII, which is negatively affected by photooxidative stress. We hypothesized that $F_v/F_m$ could be negatively impacted under conditions in which the mutants grew more slowly. The measurement of $F_v/F_m$ was performed always at the same time of the day, 1 d before the shift and 3 h after the shift to

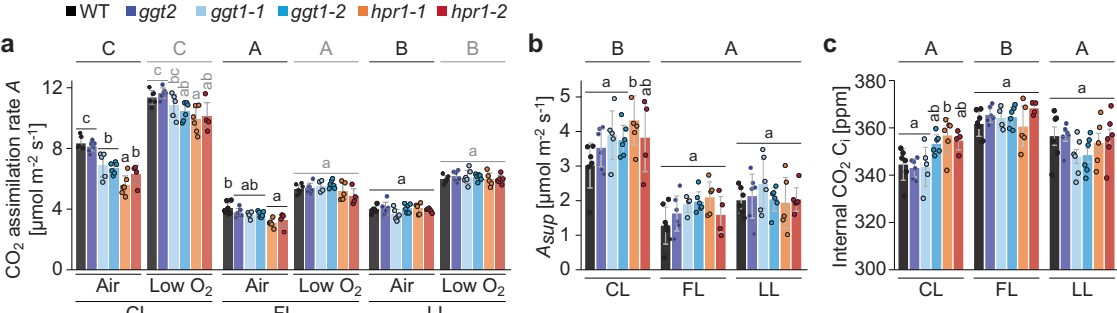

**Fig. 2 | Rubisco oxygenation rates of the different conditions increase with assimilation rates and are highest in control light. a** Plants were grown in fluctuating light (FL: 1 min 700 μmol photons m$^{-2}$ s$^{-1}$, 4 min 70 μmol photons m$^{-2}$ s$^{-1}$) in such way that they had similar biomass at the same time (WT and *ggt2* for 39 days and *ggt1* and *hpr1* for at least 50 days). Then, at this time point, all genotypes were shifted to either control light (CL: 200 μmol photons m$^{-2}$ s$^{-1}$), low light (LL: 90 μmol photons m$^{-2}$ s$^{-1}$) or kept in FL for at least 5 days. This time frame was chosen to allow for complete acclimation of plants to the new light condition. Gas exchange measurements were performed over ~ 45 min in growth light with air or low oxygen (21% or 2% O$_2$, respectively). CO$_2$ assimilation rate (*A*) was averaged over the last two light fluctuations (FL) or the final 1.5 min at the end of the measurement (CL and LL). Plants were grown and analyzed in a randomized fashion to avoid position and time of the day effects. Statistically significant differences were determined for each gas treatment independently. Capital letters indicate significant differences between light conditions and lower-case letters between genotypes within one light environment (air: black, low O$_2$: grey) as determined via two-way ANOVA and post-hoc Tukey multiple comparison test. **b** O$_2$-suppressed CO$_2$ assimilation rate (*A*$_{sup}$) was calculated as the difference between *A* in air and *A* in low O$_2$ as in panel b. **c** Internal CO$_2$ concentration C$_i$ in air was determined from gas exchange data as in panel b averaged over the last two light fluctuations (FL) or the final 1.5 min at the end of the measurement (CL and LL). **b, c** Capital letters indicate significant differences between light conditions and lower-case letters between genotypes within one light condition as determined via two-way ANOVA and post-hoc Tukey multiple comparison test with p < 0.05. **ac** Averages of n = 4 (CL: *hpr1-2*; FL: *ggt1-1*, *hpr1-2*), n = 5 (CL: *ggt1-1*; FL: *hpr1-1*; LL: *hpr1-1*), n = 6 (CL: WT, *ggt2*, *ggt1-2*, *hpr1-1*; FL: *ggt2*, *ggt1-2*; LL: WT, *ggt2*, *ggt1-1*, *ggt1-2*, *hpr1-2*) and n = 7 (FL: WT) ± standard deviation are shown. Complete statistical analysis and averaged CO$_2$ assimilation and C$_i$ traces can be found in Supplementary Data 1 and Supplementary Figs. 5, 6.

follow the immediate response and 1 d, 3 d, and 5 d after the shift to observe long-term acclimation effects on F$_v$/F$_m$ (Fig. 3, Supplementary Figs. 7, 8).

Whole rosettes of the different genotypes had comparable F$_v$/F$_m$ before the shift (t$_0$, Fig. 3). After shift to CL, F$_v$/F$_m$ slightly rose in WT and *ggt2*, significantly only for the latter after 5 d (Fig. 3a, b). In all PR mutants instead, F$_v$/F$_m$ significantly decreased after the shift. The fastest response was found in *gldt1-1*, in which F$_v$/F$_m$ dropped significantly already after 3 h. For *ggt1* and *hpr1*, a significant decrease was observed after 1 day. F$_v$/F$_m$ subsequently increased again in *gldt1-1* and *ggt1* lines, while remaining low in *hpr1* (Fig. 3b). The shift from FL to CL led to spatial differences in the F$_v$/F$_m$ of PR mutant rosettes. For *hpr1* and *gldt1-1*, F$_v$/F$_m$ showed decreased values after the shift only in the older leaves. *ggt1* instead developed a patchy low F$_v$/F$_m$ phenotype in response to the shift, which colocalized with chlorotic spots (Supplementary Fig. 7). The two distinct low F$_v$/F$_m$ phenotypes, specific for either *hpr1*/*gldt1-1* or *ggt1* mutants, were also present in mature plants grown in CL from seed to rosette and could be reversed by shifting the CL-grown plants to FL (Supplementary Fig. 9).

Together, this supports that the decrease in F$_v$/F$_m$ is caused by the CL growth condition. In addition to the decrease in F$_v$/F$_m$, PR mutant plants displayed higher levels of the different isozymes of the ROS-scavenging ascorbate peroxidase specifically in CL as compared to WT (Supplementary Fig. 10, Supplementary Data 2, 3). This result supports that PR mutant plants experience more photooxidative stress in CL than in FL. Protein levels of GGT1/2 and HPR1 were unaffected by growth light (Supplementary Fig. 10, Supplementary Data 2, 3).

A shift from FL to LL led to an increase in F$_v$/F$_m$ in all genotypes (Fig. 3c, d). However, after 1 day in LL, similarly to CL, *ggt1* developed patches with lower F$_v$/F$_m$, which persisted throughout the 5-day measuring period (Fig. 3c, Supplementary Fig. 8). *hpr1* and *gldt1-1* did not show any F$_v$/F$_m$ heterogeneity between leaves of different developmental stages in LL.

This analysis shows that the strong growth retardation of *ggt1* in LL and CL coincides with the development of low F$_v$/F$_m$ patches. The growth retardation of the *hpr1* and *gldt1-1* mutants in CL coincides with low F$_v$/F$_m$ in older leaves.

## FL to CL shift induces genotype-specific metabolic reprogramming

The PR pathway is deeply embedded in primary metabolism, with metabolites being diverted into amino acid production, the TCA cycle or other metabolic pathways[20] (Fig. 4). Differences in PR metabolite accumulation had already been described for the two "mild" PR mutants *ggt1*[49] and *hpr1*[50], but not when grown side-by-side and in dynamic growth light. We hypothesized that a differential engagement of metabolic pathways as a function of growth light in both mutants may be the cause for the observed differences in growth and particularly F$_v$/F$_m$ between *ggt1* and *hpr1*. Because localized F$_v$/F$_m$ patches were most pronounced in *ggt1* within the first few days after shift from FL to CL (Fig. 3a, b), this shift was further analyzed to pinpoint light condition-dependent differences in primary metabolism (Fig. 4, Supplementary Fig. 11, Supplementary Data 4). To account for potential developmental effects during the light shift, rosette material was additionally sampled from all genotypes shifted from CL to FL or grown only in FL or CL (Supplementary Figs. 12–14, Supplementary Data 4).

Overall, the shift from FL to CL was accompanied by strong changes in the levels of primary metabolites with pronounced differences between the genotypes. While total carbohydrate levels were comparable between all genotypes in FL, the shift induced their accumulation particularly in WT and *ggt2* and much less in both *ggt1* and *hpr1* (Fig. 4, Supplementary Fig. 11, Supplementary Data 5). Total levels of free amino acid were less affected by the light shift (Fig. 4, Supplementary Fig. 11). While *ggt* mutants had WT-like amino acid content over all sampled time points, *hpr1* displayed elevated levels throughout the measurements (Fig. 4, Supplementary Data 5).

In response to the light shift, most measured metabolites gradually changed until reaching new CL steady-state levels. For the PR metabolites glycine and glycerate, we observed a different behavior: These metabolites underwent strong transient changes in WT that were most pronounced at 1 day after the shift and then reversed, with glycine transiently spiking and glycerate dropping. This finding may be explained by the transition from FL to CL leading to a transient increase in PR flux.

PR metabolite accumulation patterns were distinctly different in both PR mutants compared with WT: In *hpr1*, all four quantifiable PR

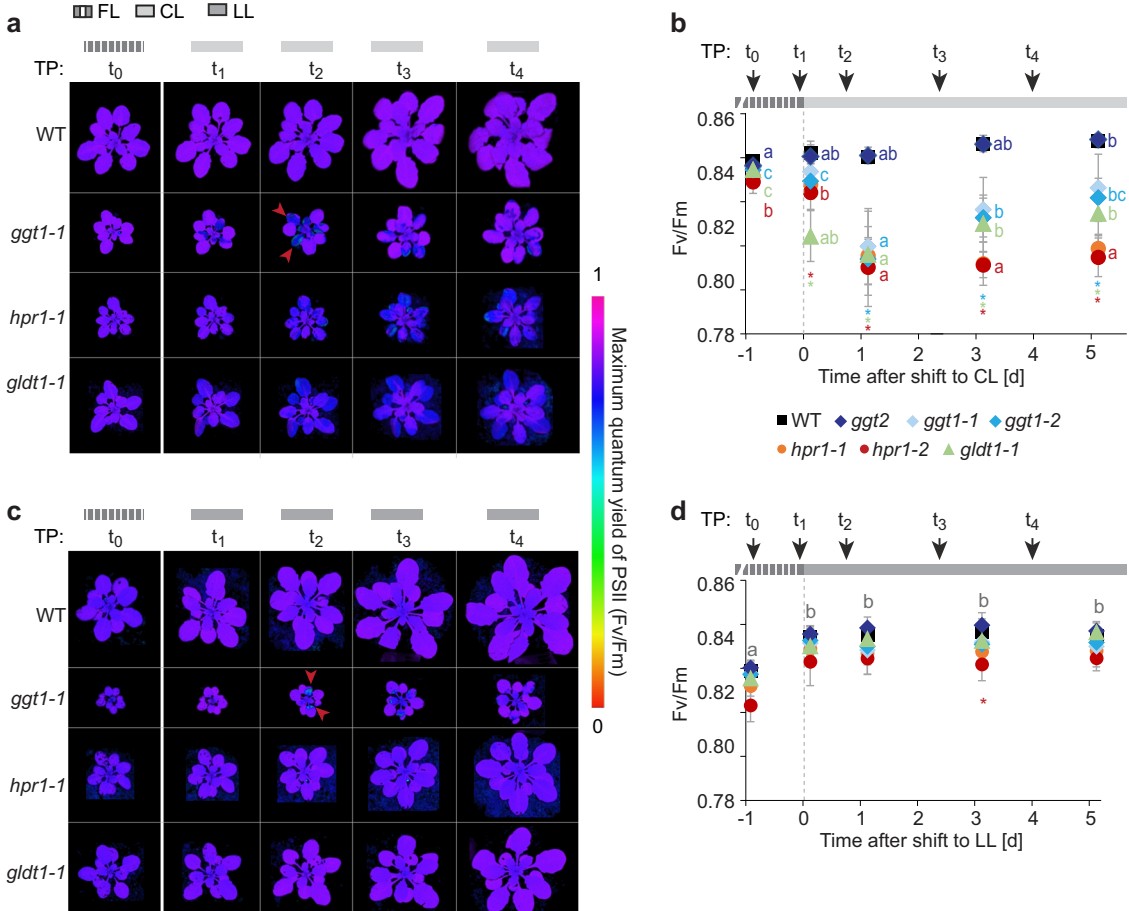

**Fig. 3 | Distinct differences in $F_v/F_m$ between photorespiratory mutants after shift from fluctuating to control or low light. a–d** Measurements of the maximum quantum efficiency of PSII ($F_v/F_m$) of WT, *ggt1-1*, *hpr1-1* and *gldt1–1* grown under fluctuating light (FL: 1 min 700 µmol photons $m^{-2} s^{-1}$, 4 min 70 µmol photons $m^{-2} s^{-1}$; time point $t_0$) and 3 h ($t_1$), 27 h ($t_2$), 75 h ($t_3$) and 123 h ($t_4$) after the shift to control light (CL: 200 µmol photons $m^{-2} s^{-1}$; a) or low light (LL: 90 µmol photons $m^{-2} s^{-1}$; c). **a, c** Representative false color images of $F_v/F_m$ as indicated by the false color bar on the right. Red arrows in $t_2$ point to patchy low $F_v/F_m$ found in the *ggt1* mutant only. Images of all time points can be found in Supplementary Figs. 7, 8. **b, d** Average $F_v/F_m$ of the experiment as in **a** (**b**) or **c** (**d**). Averages of n = 10 for all genotypes except for n = 7 for *hpr1-1* (b, $t_4$), n = 8 for *hpr1-2* (b, $t_4$), n = 9 for *hpr1-1* (b, $t_0$-$t_3$), *gldt1-1* (b, $t_0$-$t_4$) and *ggt1-2* (b, $t_4$) and n = 11 for *hpr1-2* (b, $t_0$-$t_3$) ± standard deviation are shown. Asterisks in the corresponding color indicate significant differences between mutants and WT within one time point (TP) and different lowercase letters in the corresponding color (or in grey for all plants) significant difference between time points within one genotype determined via two-way ANOVA and post-hoc Tukey multiple comparison test with $p < 0.05$. To exclude background signal due to algal growth on the soil, the outline of each plant as in a and c was selected manually to generate the average $F_v/F_m$ shown in panels **b** and **d**.

metabolites significantly increased after the shift (i.e., glycine after 3 h, serine and glycolate after 1 d, and glycerate after 3 d). This would be consistent with higher 2PG production and a low capacity for PR flux in the absence of HPR1. Serine was higher compared with WT already before the shift (Fig. 4, Supplementary Fig. 11, Supplementary Data 5). The strongest difference that was observed for *ggt1* was a dampened increase in glycine from day 1 after the shift onwards. The *ggt2* mutants also showed a delayed increase of glycine after the shift, not to the same extent as *ggt1*, but significant compared with WT. Glycine is the product of the photorespiratory GGT reaction, suggesting that GGT2 confers the same activity as GGT1 in PR and contributes to the transient increase in glycine after the shift in WT (Fig. 4). Our data thus support a function for this enzyme in response to increased rates of PR, when plants experience a transition to conditions with higher rates of Rubisco oxygenation reaction, such as the shift from FL to CL.

Metabolites of the TCA cycle were affected by genotype and light condition and respective interactions (Fig. 4). Most notably, we found an increase in fumarate and malate in all genotypes after the FL to CL shift, with *ggt1* containing lower fumarate levels than WT across all sampling time points. Additionally, *ggt1* had increased citrate levels at most sampling time points (Fig. 4, Supplementary Figs. 12–14). Generally,

levels of pyruvate and TCA cycle metabolites (except for citrate) were higher in *hpr1* than in WT after the shift to CL. TCA cycle metabolites of *ggt2* were WT-like, independently of the light environment.

Together, the results of the metabolite analysis during the shift from FL to CL reveal common as well as opposing differences when comparing *ggt1* and *hpr1* with WT. Particularly metabolites of PR and the TCA cycle showed contrasting accumulation patterns with general increases in *hpr1* and some decreases in *ggt1*. Most metabolites of the TCA cycle accumulated to high levels in *hpr1* after the shift, while in *ggt1* fumarate was decreased at all time points. All PR metabolites accumulated in *hpr1* in response to the light shift, while both glycine and serine decreased in *ggt1* compared with WT. Glycolate is an upstream metabolite of both the HPR1 and GGT1 reactions, but only accumulates to high levels in *hpr1* and not *ggt1* after the shift. One plausible explanation for this is that alternative glycolate processing reactions are upregulated in the absence of GGT1.

**Metabolic modeling predicts genotype- and growth condition-dependent flux modes of PR and TCA cycle**

To obtain an integrative understanding of how the two different enzymatic defects of *hpr1* and *ggt1* affect metabolism under CL and FL,

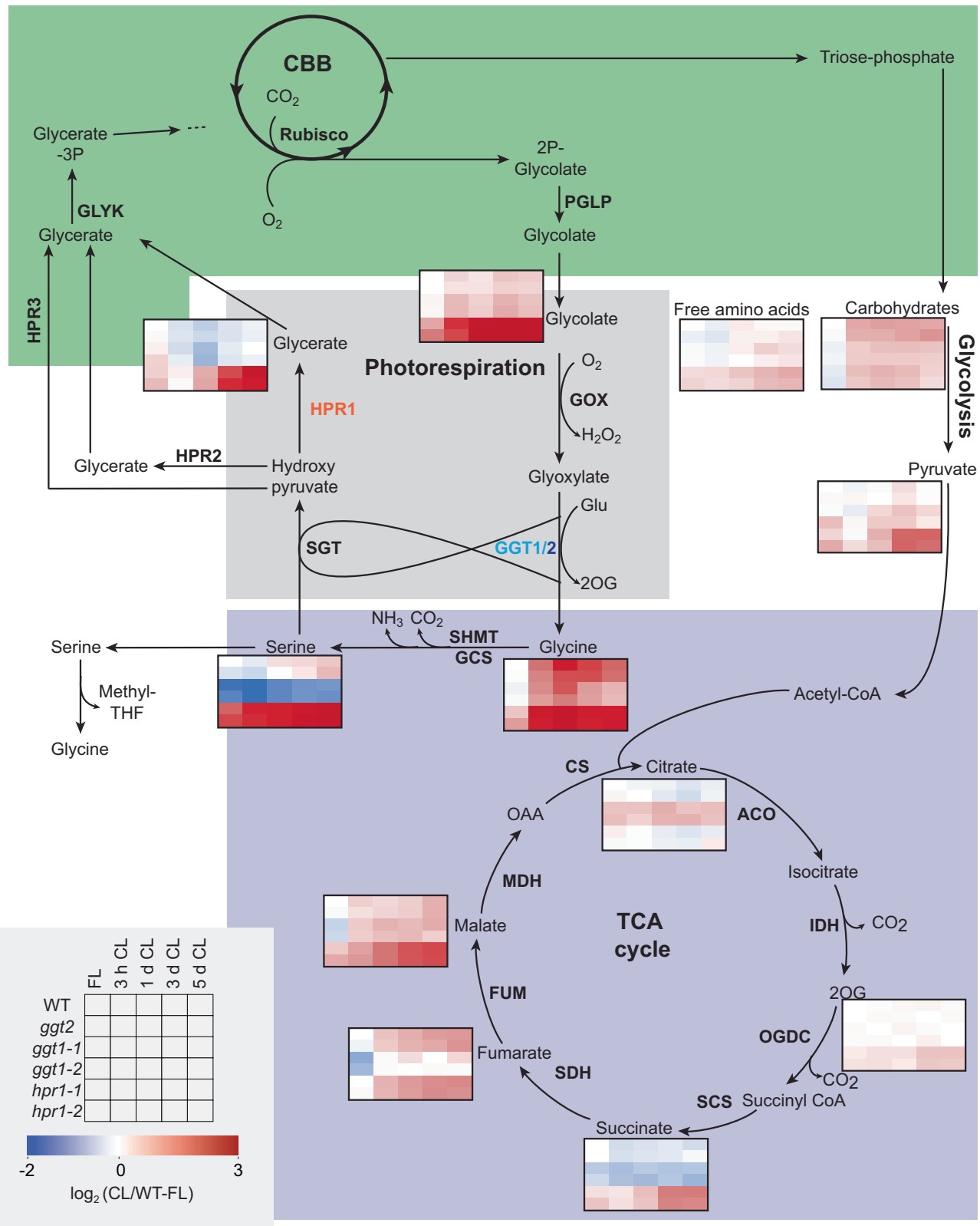

we employed thermodynamics-based metabolic flux analysis (TMFA) using the core model of Arabidopsis primary metabolism[51] (Fig. 5). The metabolite data from steady state FL and CL as well as the respective growth and $CO_2$ assimilation rates served as input to predict the rates of metabolic reactions under each condition.

The model predicted higher flux through the Rubisco oxygenase reaction in CL compared with FL (Fig. 5a, Supplementary Data 6,

7). Accordingly, WT flux through two of the initial PR enzymes glycolate oxidase (GOX) and GGT1 in CL exceeded that of FL (Fig. 5a, b). Glycine processing by the reactions of the mitochondria-localized serine hydroxymethyl transferase (SHMT) and the glycine cleavage system (GCS) was higher in FL than in CL (Fig. 5c), resulting in increased pool sizes of glycine in the latter condition (Supplementary Data 4).

**Fig. 4 | Photorespiratory mutants show distinct differences in metabolic restructuring during light acclimation.** WT, *ggt2* and two mutant alleles of each *ggt1* and *hpr1* were shifted from fluctuating light (FL: 1 min 700 μmol photons m$^{-2}$ s$^{-1}$, 4 min 70 μmol photons m$^{-2}$ s$^{-1}$) to control light (CL: 200 μmol photons m$^{-2}$ s$^{-1}$) as in Fig. 3a, b. Heatmaps of log$_2$ transformed metabolite averages normalized to WT in FL (representation scheme shown in the bottom left corner of the figure) are shown for total carbohydrate and free amino acid levels, photorespiratory metabolites and tricarboxylic acid (TCA) cycle metabolites in the chloroplast (green), peroxisome (grey), mitochondrion (purple) and cytosol (white). The results show that carbohydrates are less induced in both mutants compared with WT in response to the shift, while some metabolites of the TCA and PR cycles are more strongly induced in *hpr1* and reduced in *ggt1* after the shift. Total carbohydrates and free amino acids were calculated from replicates for which the main constituent (sucrose and serine and glutamine, respectively) could be detected. More

data on metabolite analyses can be found in Supplementary Figs. 11–14 and the complete data sets in Supplementary Data 4. Averages of $n = 3$-4 replicates are shown. The exact number of replicates per group can be found in Supplementary Data 4. Statistical analysis can be found in Supplementary Data 5. CBB – Calvin-Benson-Bassham cycle; Enzymes: PGLP – 2-phosphoglycolate phosphatase; GOX – Glycolate oxidase; GGT1/2 – Glutamate:glyoxylate aminotransferase 1/2; SHMT – Serine hydroxymethyltransferase; GCS – Glycine cleavage system; SGT – Serine:glyoxylate aminotransferase; HPR1-3 – Hydroxypyruvate reductase 1-3; GLYK – D-glycerate 3-kinase. ACO – Aconitase; IDH – Isocitrate dehydrogenase; OGDC – 2-oxoglutarate dehydrogenase complex; SCS – Succinyl-CoA synthetase; SDH – Succinate dehydrogenase; FUM – Fumarase; MDH – Malate dehydrogenase; CS – Citrate synthase; Metabolites: 2P-Glycolate – 2-phosphoglycolate; Glycerate-3P – Glycerate-3-phosphate; Ribulose-1,5-bisP – Ribulose-1,5-bisphophate; OAA – Oxaloacetate.

Most PR reactions were slightly decreased in *hpr1* compared with WT under both conditions (Fig. 5a–c), except for flux from glycine to serine via SHMT and GCS, which was increased in CL. The model could not realistically discriminate between HPR1-3, as it predicted the highest activity flux through HPR2, which represents a minor route for hydroxypyruvate processing as compared to HPR1[37]. Low PR flux downstream of SHMT and GCS reactions was predicted for WT and *hpr1* under both growth light conditions.

In all growth conditions and genotypes, the TCA cycle was predicted to run in a non-cyclic mode (Fig. 5c), as reviewed in ref. 52. Isocitrate dehydrogenase (IDH) was always predicted to be inactive as well as the two enzymes downstream of the IDH that confer flux from 2-oxoglutarat to succinate. These two enzymes showed low activity only in *ggt1* from FL. In WT, flux from succinate to isocitrate was higher in FL than in CL, except for malate dehydrogenase (MDH), which showed comparable activities under both conditions. *hpr1* provided higher flux through these TCA cycle reactions compared with WT in CL (Supplementary Data 6).

### Increased mitochondrial ATP production in CL grown *ggt1* and *hpr1* involves distinct reductive pathways

The model predicted increased total ATP and NADH flux sums for both PR mutants compared with WT, particularly in CL, which was mainly due to increases in mitochondrial metabolism (Supplementary Data 6). The enzymes contributing to the increase in mitochondrial NADH production compared with WT differed between *ggt1* and *hpr1*: While the model predicted increased mitochondrial MDH and combined SHMT and GCS activities in *hpr1*, activities of two glutamate dehydrogenase were higher in *ggt1* (Fig. 5c; Supplementary Data 6). These two enzymes cycle between glutamate and 2-oxoglutarate, resulting in the net conversion of NADPH to NADH. The modeled increase in ATP production compared with WT resulted from higher rates of the mitochondrial respiratory electron chain using the surplus of NADH released in the mitochondria of the mutants (Supplementary Data 6).

### Glycolate processing is rerouted in *ggt1* leading to high hydrogen peroxide release in the chloroplast under non-fluctuating light conditions

The lack of glycolate accumulation in *ggt1* suggests that this PR metabolite is processed by alternative pathways in the absence of GGT1. Consistent with this, the model showed increased glyoxylate flux through the serine:glyoxylate aminotransferase (SGT) reaction and higher activities of downstream reactions in *ggt1* compared with WT under both conditions, but particularly in FL with approximately 3-times the flux of CL (Fig. 5b, c, Supplementary Data 6). In CL, the major activity for glycolate processing in *ggt1* was predicted to be located in the chloroplast. The Arabidopsis core model contains a chloroplast localized GOX and GGT enzymatic activity and predicted that flux through this pathway was negligible across genotypes and both conditions except for *ggt1* from CL. In these plants, both

chloroplast GOX and GGT reactions were specifically upregulated, while the peroxisomal GOX was downregulated (Fig. 5a). In *ggt1* from CL, H$_2$O$_2$ production shifted along with the GOX enzyme activity from the peroxisome to the chloroplast.

## Discussion

We set out to test the hypothesis that a high competence for PR is particularly relevant for plant performance under strongly fluctuating light. It was proposed that under such dynamic light conditions, high PR flux is needed to respond to increased rates of 2PG production and spikes in metabolic energy, which result from the kinetic inefficiencies of CBB cycle activation and stomatal conductance. Our combined results from analyzing different PR mutants do not support the hypothesis of higher relevance of PR in dynamic light. Instead, they reveal that growth of PR deficient mutants was positively affected by strongly fluctuating light (Fig. 1).

All analyzed genotypes displayed higher rates of photosynthesis under CL compared with FL (Fig. 2a), with both conditions providing the same average light intensity over the day. Thus, our results reinforce the notion that photosynthesis cannot use fluctuating light as efficiently as constantly supplied light. Besides CBB cycle activation and stomatal conductance[21,28], also photoprotective mechanisms[53] were shown to limit the efficiency of light utilization for photosynthesis in fluctuating light. The Rubisco carboxylation/oxygenation ratios were not greatly impacted by the growth light in our experiments (Fig. 2). This suggests that one explanation for the improved growth of PR mutants in FL lies in the overall lower rates of 2PG production under this condition (Fig. 6). Consequently, PR flux in the *ggt1* and *hpr1* mutants would be sufficient to avoid negative effects on photosynthesis by 2PG accumulation. The *hpr1* mutant had comparable growth rates with WT not only in FL, but also in LL. WT photosynthetic rates were similar between these two conditions, suggesting that *hpr1* grew well in LL, because as in FL, residual PR flux was sufficient to avoid detrimental 2PG accumulation.

However, the *ggt1* mutant growth phenotypes could not be explained solely by the rates of photosynthesis and 2PG production under different conditions. The growth of *ggt1* mutants was also strongly impaired in LL, despite plants displaying only half of the photosynthetic activity compared with CL and similar rates as in FL (Figs. 1, 2).

The *ggt1* mutant rosettes formed low $F_v/F_m$ patches when grown under non-fluctuating light conditions, which co-localized with chlorotic spots (Fig. 3). This growth light-dependent phenotype can also be found in the GO mutants[54,55]. In addition to the normal peroxisomal GOX, these plants contain an additional GOX targeted to the chloroplast. The GO mutants produce high levels of H$_2$O$_2$ in the chloroplast, which was hypothesized to cause the variegated phenotype[55]. Alternative processing of glycolate in isolated chloroplasts was demonstrated by the release of CO$_2$ from radioactively labelled glycolate[7]. Our metabolic flux modeling supported that alternative glycolate

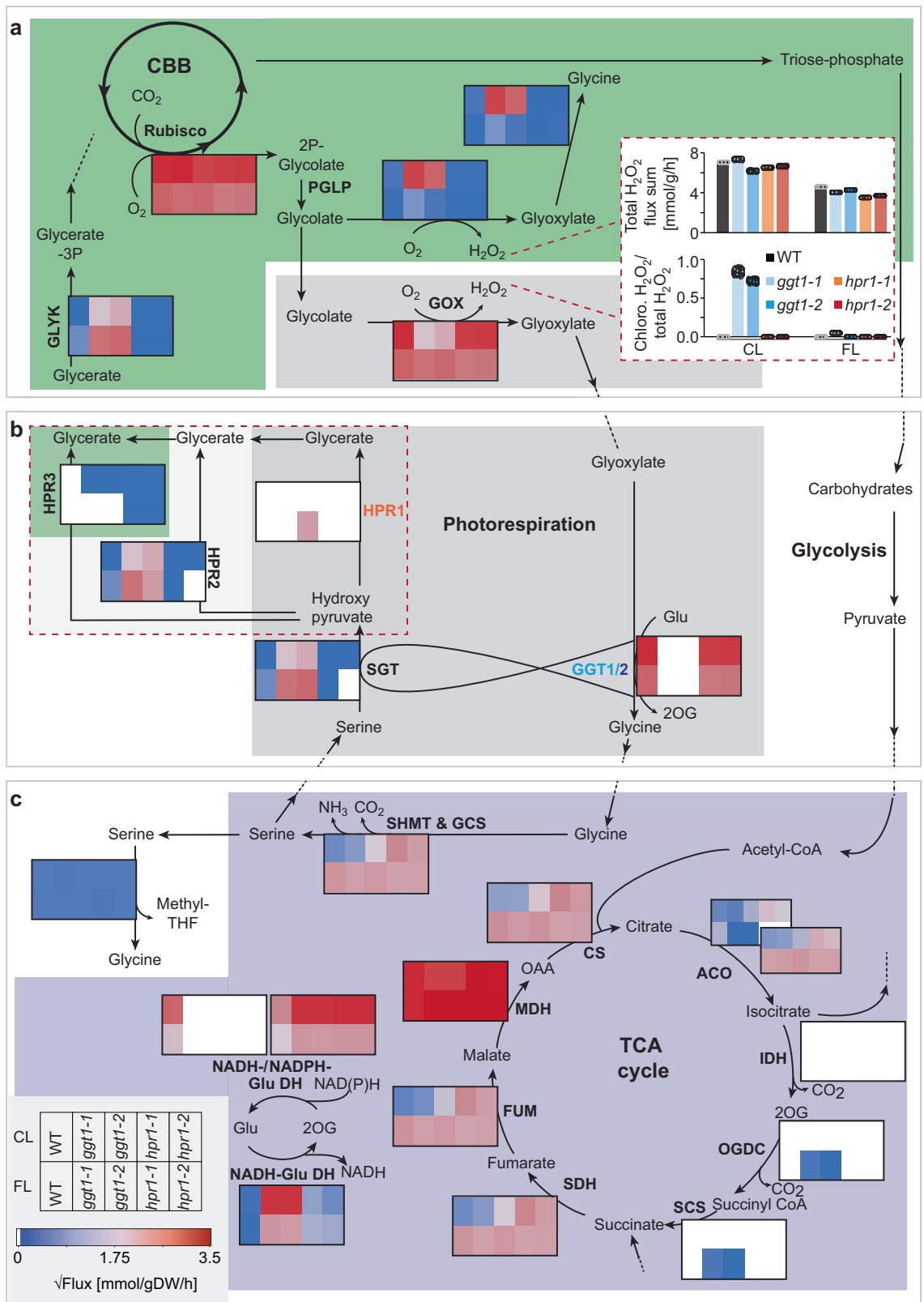

processing occurs in the chloroplasts of *ggt1* when grown in CL. This reaction results in increased release of $H_2O_2$ in this organelle (Fig. 5).

The presence of glycolate processing in the chloroplast raises the question towards the physiological function of such pathway. Here, we propose that a chloroplast-localized glycolate decarboxylation pathway may be beneficial for plant photosynthesis during light fluctuations (Fig. 6). A chloroplast-localized reaction could metabolize 2PG

spikes quickly and release $CO_2$ in close vicinity to Rubisco. 2PG spikes are expected to occur after low to high light transitions, which induce a sudden decrease in $C_i$ and thus increase in the Rubisco oxygenation reaction[56,57]. The *ggt1* mutants showed the variegated phenotype also under LL, in which photosynthetic rates were comparable to FL. Thus, we propose in line with previous hypotheses[54], that under FL specific chloroplast-localized scavenging systems (i.e., catalases) are

**Fig. 5 | Predicted re-routing of metabolite flux in response to light patterns and photorespiratory perturbations.** The AraCoreTFA model was parametrized with $CO_2$ assimilation and relative growth rates and metabolite concentrations to predict flux distributions for the Col-0 wild type (WT) and photorespiratory mutants *ggt1-1, ggt1-2, hpr1-1, hpr1-2* in control light (CL: 200 µmol photons $m^{-2}$ $s^{-1}$) and fluctuating light (FL: 1 min 700 µmol photons $m^{-2}$ $s^{-1}$, 4 min 70 µmol photons $m^{-2}$ $s^{-1}$). **a–c** The heatmaps display the square root of flux (representation scheme shown in the bottom left corner of the figure) through photorespiration (**a–c**) and TCA cycle (**c**) and other selected reactions in the chloroplast (green), peroxisome (grey), mitochondrion (purple) and cytosol (white). The bar graph inlay in panel a shows total $H_2O_2$ flux sums (top) and relative chloroplast (chloro.) per total $H_2O_2$ values (bottom), pointing to high release of $H_2O_2$ in the chloroplasts of *ggt1* mutants only in CL. Error bars indicate standard deviation. The flux sum through a metabolite is one half of the sum of absolute value of the contributions of different reactions that alter the concentration of the metabolite (i.e., $0.5 \cdot \sum_{j} |N_{ij}v_j|$). The dashed box in panel b encloses all known enzymatic reactions that contribute to the conversion of hydroxypyruvate to glycerate. For *hpr1* simulations, HPR1 (orange) was removed from the model and for *ggt1*, GGT1 (blue). The white color indicates no flux. Note that reducing equivalents were only added to glutamate dehydrogenase (Glu DH) reactions. Averages of $n = 100$ replicates are displayed. CBB – Calvin-Benson-Bassham cycle; Enyzmes: PGLP – 2-phosphoglycolate phosphatase; GOX – Glycolate oxidase; GGT1/2 – Glutamate:glyoxylate aminotransferase 1/2; SHMT – Serine hydroxymethyltransferase; GCS – Glycine cleavage system; SGT – Serine:glyoxylate aminotransferase; HPR1-3 – Hydroxypyruvate reductase 1-3; GLYK – D-glycerate 3-kinase. ACO – Aconitase; IDH – Isocitrate dehydrogenase; OGDC – 2-oxoglutarate dehydrogenase complex; SCS – Succinyl-CoA synthetase; SDH – Succinate dehydrogenase; FUM – Fumarase; MDH – Malate dehydrogenase; CS – Citrate synthase; Metabolites: 2P-Glycolate – 2-phosphoglycolate; Glycerate-3P – Glycerate-3-phosphate; Glu – Glutamate; Ribulose-1,5-bisP – Ribulose-1,5-bisphosphate; OAA – Oxaloacetate.

upregulated that avoid the build-up of $H_2O_2$ in this organelle and thus mitigate negative effects on the photosynthetic machinery (Fig. 6c, d, hypothetical ROS scavenger depicted as blue circle). Higher rates of chloroplast glycolate processing in CL together with additional FL-specific upregulation of $H_2O_2$ scavenging systems may explain the observed light condition-dependent phenotypes of *ggt1*. Synthetic PR bypasses within the chloroplast that introduced glycolate converting enzymes into this organelle were shown, under some conditions, to increase biomass of field-grown tobacco[12]. Uncovering the enzymes responsible for the endogenous bypass revealed in this study and the signal(s) that control its activity may provide avenues to improved crop performance. As PR metabolism is strongly conserved amongst higher plants (reviewed in ref. 58), such strategies may be applicable to most $C_3$ crops. With this perspective, it still remains to be determined whether fluctuating light causes similar re-routing of PR and other metabolic pathways in crops and to which extent natural dynamic light conditions differ in their effects.

Alternatively, the growth benefits of the *ggt1* mutant in FL may exclusively link back to glycolate processing only occurring in the peroxisome and not in the chloroplast under this growth condition (Fig. 5). The simulated FL-specific high flux through the SGT reaction places PR-linked $H_2O_2$ production back into the peroxisome, with less negative effects on chloroplast-localized photosynthetic processes.

When grown in CL, the *hpr1* mutants showed lower $F_v/F_m$ and yellowing of older leaves (Fig. 3). This phenotype is indicative for the onset of accelerated senescence[59]. The accumulation of free amino acids in the entire rosette of *hpr1* under this condition supported this notion, as proteins are degraded in senescing leaves to free amino acids for younger leaf tissue[60]. Additionally, mitochondria take on a central role in cellular energy production during senescence (reviewed in ref. 61). NADH is produced in the mitochondria and then used for ATP production. Multiple oxidative reactions resulting in the release of NADH in the mitochondria were predicted to increase in *hpr1* compared with WT specifically in CL, leading to elevated mitochondrial NADH and ATP flux sums (Fig. 5, Supplementary Data 6). These include the conversion of glycine to serine via the SHMT and GCS and higher flux from succinate to citrate, which are reactions associated with the break-down of amino acids (reviewed in ref. 62).

We used a metabolic model of a growing leaf; hence we assume maximization of the RGR for the WT under constant light as the primary objective for TMFA optimization. For all other considered scenarios (i.e., modeling of mutants and FL condition), we did not optimize the RGR, because growth in these cases may not present the primary objective. Instead, we fixed the RGRs, which were calculated from the measured dry weights, relative to the WT-RGR of CL. This procedure should account for differences in primary objectives given by the fluctuating light or PR-enzyme deficiencies.

For WT, the model predicted a minor role for the recycling of CBB cycle intermediates by flux through the PR core reactions[63] (Fig. 5).

Instead, it revealed a key function for PR in the production of glycine particularly in CL and serine particularly in FL. Predicted PR flux towards these amino acids was high (i.e., until glycine in CL and serine in FL), with much lower activities of downstream reactions. Both, glycine and serine are considered "safe", non-toxic PR intermediates, because high levels do not negatively affect plant metabolism and growth[57]. Along with higher mitochondrial flux from glycine to serine, the active enzymes of the TCA cycle were also predicted to have increased rates in FL compared with CL (Fig. 5), accompanied by higher mitochondrial ATP flux sums via increased activity of the respiratory electron transport chain (Fig. 6a, b, Supplementary Data 6). This adjustment of WT metabolism under fluctuating light may provide extra ATP for the plant to cope with increased photooxidative damage. Higher levels of photooxidative damage in FL compared with CL were indicated by the lower $F_v/F_m$ in this condition (Fig. 3).

A second flux sampling approach[64], providing a wider coverage of the flux space, showed robustness of the conclusions (Supplementary methods, Supplementary Fig. 16).

Together, our analyses of genetic PR perturbations under different light conditions reveal that plant growth in fluctuating light buffers against PR perturbations in two ways, by (i) lowering photosynthetic rates and (ii) beneficial re-routing of metabolite fluxes. This discovery mandates a shift in our perception of fluctuating light as a stress. Instead, our data point towards growth in fluctuating light providing plants with additional buffering mechanisms allowing them to better cope with perturbations in metabolism.

## Methods

### Plant material and growth conditions

*Arabidopsis thaliana* plants (Columbia-0) WT plants and T-DNA insertion lines of the genes *HPR1* (AT1G68010; *hpr1-1*: Salk 067724; *hpr1-2*: Salk 143584)[36], *GGT1* (AT1G23310; *ggt1-1*, Gabi-Kat 649H07; *ggt1-2*, Gabi-Kat 847E07)[39,49] and knock down of GCS T-protein (At1g11860; *gldt1-1*: WiscDsLox366A11_085; accumulates only 5% of WT-protein level)[41] as well as overexpressors of the GCS L-protein from *Pisum sativum* (X63464; *PsL-L2* in ref. 46, ~250% of enzyme activity compared with WT) and PGLP from *Arabidopsis thaliana* (AT5G36700; *PGLP-O1* in ref. 3, ~150% of enzyme activity compared with WT), were used in the study. A T-DNA insertion line in *GGT2* (AT1G70580; *ggt2*: Gabi-Kat 526E07)[65] was obtained from NASC and confirmed using the primers: 5'-CTGGCCAGTGTCTTAGCGA-3' and 5'-GAGTTTGACACA-GAGTAGGACCA-3'. Plants were grown in a phytochamber (BBC York, Mannheim, Germany) on soil (MPG horticultural substrate, Stender, Schermbeck, Germany, with 1 g/l Osmocote start) in 6 cm pots under 12 h/ 12 h light/ dark cycles at 20/ 16 °C and 60/ 75% relative humidity. The growth light was either provided at constant intensity throughout the light period (low light, LL: 90 µmol photons $m^{-2}$ $s^{-1}$; control light, CL: 200 µmol photons $m^{-2}$ $s^{-1}$ or 250 µmol photons $m^{-2}$ $s^{-1}$) or in a fluctuating manner (fluctuating light; FL: 4 min 70 or 90 µmol photons

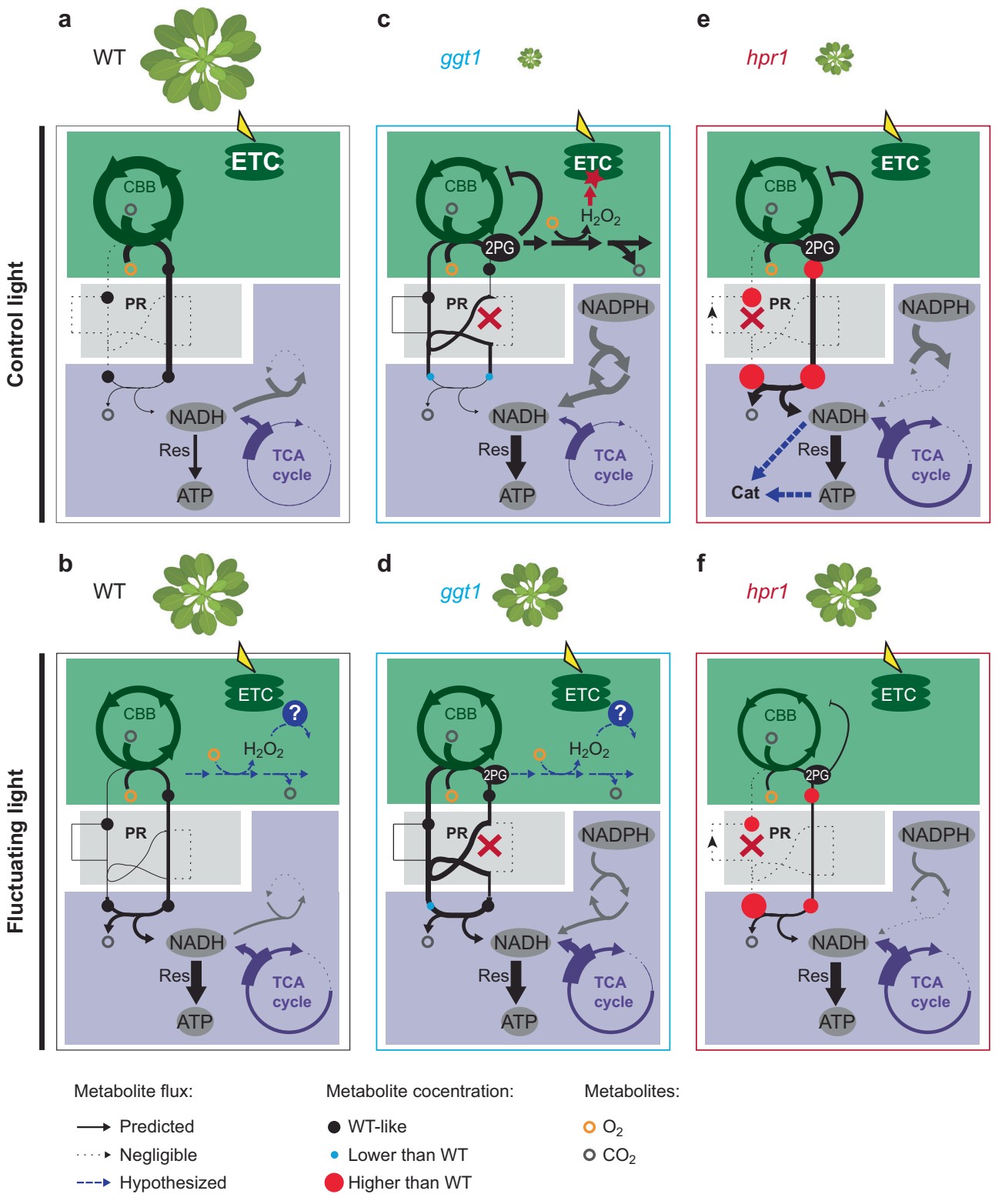

Metabolite flux:
→ Predicted
⋯▸ Negligible
--▸ Hypothesized

Metabolite cocentration:
● WT-like
● Lower than WT
● Higher than WT

Metabolites:
○ O₂
○ CO₂

m⁻² s⁻¹ and 1 min 700 or 900 μmol photons m⁻² s⁻¹, respectively), supplied by red, blue and white LEDs (Roschwege, Greifenstein, Germany, blue: 450 nm, red: 660 nm). CL and FL regimes with lower light intensities were used if not indicated differently. White LEDs accounted for 60% of the supplied light intensity, red and blue LEDs combined for 40% at the same ratio for all conditions. For all light shift experiments plants were grown for 35 d after sowing (d.a.s.; FL) or 29 d.a.s. (CL). On the day of the light shift, the light regime was changed at 3 h into the light period. If not indicated differently, measurements and

samples were taken at 6 h into the light period on each of the sampling days (21 h before and 3 h, 27 h, 75 h, 123 h after the light shift). Dry weight was determined from whole rosettes oven-dried at 67 °C for 7 d.

**Transient expression of GGT2-YFP fusions in *Nicotiana benthamiana* leaves**
The coding sequences for *GGT2* and YFP were amplified from cDNA or the $cTP_{KEA3}$ − YFP constructs[66], respectively, and assembled into pEarleyGate100[67] linearized with *Xba*I and *Xho*I restriction enzymes

**Fig. 6 | Knocking out the reactions catalyzed by HPR1 and GGT1 causes distinct metabolic restructuring during light acclimation. a–f** Proposed model of primary metabolism of WT (**a, b**), *ggt1* (**c, d**) and *hpr1* (**e, f**) grown in control light (CL: 200 μmol photons m$^{-2}$ s$^{-1}$; **a, c, e**) or fluctuating light (FL: 1 min 700 μmol photons m$^{-2}$ s$^{-1}$, 4 min 70 μmol photons m$^{-2}$ s$^{-1}$; **b, d, f**). Different growth phenotypes are indicated by the different sizes of Arabidopsis plants above the sketch of metabolic pathways. Open circles depict oxygen (orange) or carbon dioxide (grey) and closed circles measured photorespiratory metabolites (black: WT-like, red: higher absolute level, blue: lower absolute level compared with WT). Dotted lines indicate negligible and dashed, blue lines hypothesized flux. CBB – Calvin-Benson-Bassham cycle, ETC – Electron transport chain; PR – Photorespiration; TCA cycle – Tricarboxylic acid cycle; Res – Respiration; Cat – catabolic reactions. **a, b** In WT, growth in CL leads to higher flux through most primary metabolism and faster growth compared to FL: CL provides light in steady supply and thus allows for high rates of photosynthesis. In FL, 2PG is processed to serine, while there is only a low flux from serine to glycine in CL. In both conditions, flux between serine and 3-phosphoglycerate was predicted to be low, indicating a minor function of PR to replenish CBB cycle intermediates, but rather to produce glycine and/or serine. **c, d** Growth deficit of *ggt1* mutants is alleviated in FL compared with CL (and LL), at

least partially independent of high pressure on PR metabolism. A chloroplast glycolate and glyoxylate decarboxylation pathway is hypothesized to process some glycolate, potentially all the way to $CO_2$. The proposed byproduct $H_2O_2$ is detoxified by an unknown factor (question mark in blue circle) which is upregulated in an FL-specific manner. Lower levels or absence of this factor in CL lead to photooxidative damage at the chloroplast ETC. High production of NADH in the mitochondria in both light conditions is provided via glutamine dehydrogenases (grey arrows) that lead to the net production of NADH from NADPH. **e, f** Knockout of the peroxisomal HPR1 reaction leads to a more severe growth deficit and metabolic differences to WT in CL compared with FL. CBB cycle activity together with pressure on PR are higher in CL than in FL and lead to a stronger accumulation of PR metabolites (accumulated metabolites indicated as red circles in different sizes, according to metabolite levels relative to WT), potentially also 2PG. The latter leads to a stronger inhibition of photosynthesis in CL compared to FL. Higher flux through PR towards serine and MDH activity in CL produce higher amounts of reducing equivalents, fueling respiration for ATP production. ATP and NADPH deliver the energy for catabolic reactions during leaf senescence. This figure was created with BioRender.com.

downstream of the *35S* promoter by using Gibson Assembly[68]. The plasmid is available upon request. *Agrobacterium tumefaciens* strain GV3101 transformed with the *GGT2 – YFP* construct was resuspended in induction medium (10 mM MgCl$_2$, 10 mM MES-KOH pH 5.6, 150 μM acetosyringone) to an OD600 of 0.5. After 2 h at 28 °C, the suspension was inoculated onto sections of well-watered *N. benthamiana* (tobacco) leaves by injecting into the bottom side of a punctured leaf[69]. Transfected plants were grown for 2 d in room light before detached leaves were analyzed for the localization of the YFP fluorescence signal by confocal microscopy. For microscopy, the Leica TCS SP5 instrument was used with 63x/1.4 objective and water immersion. Fluorophores were excited by using an argon laser at 514 nm, YFP fluorescence was detected between 524 and 582 nm, and chlorophyll fluorescence between 618 and 800 nm.

## Chlorophyll *a* fluorescence analysis

The MAXI IMAGING-PAM (Walz, Effeltrich, Germany) was used for chlorophyll *a* fluorescence analysis of whole rosettes. Minimal fluorescence $F_0$ and maximal fluorescence $F_m$ of 30 min dark-adapted plants were derived from the measuring light only and during application of a saturating light pulse (0.8 s), respectively. The maximum quantum efficiency of PSII, $F_v/F_m$, was calculated as $(F_m-F_0)/F_m$. Measurements were conducted 6 h into the light period. For each experiment, the same set of 7-10 plants for each genotype was measured repeatedly at the indicated time points (i.e., once per day).

## Metabolite extraction and quantification

Whole rosettes (1-4 plants pooled per replicate) were harvested 6 h into the light period and immediately frozen in liquid nitrogen 1 day before the shift in growth light (FL: 35 d.a.s., CL: 29 d.a.s.) and 3 h, 27 h, 75 h and 123 h after the shift. The material was ground using a cryogenic grinder system (Labman). Samples for metabolite profiling were extracted using a method optimized for photorespiratory metabolites[70], and analyzed by GC-MS[71]. In detail, 50 mg of frozen leaf powder was mixed with 600 μl ice-cold N,N-dimethylformamide (100%). Subsequently, 5 μl of Ribitol (0.2 mg/ml) and 400 μl H$_2$O were added and left shaking at 14000 rpm and 4 °C for 10 min. The sample was centrifuged for 10 min at 22000 x g and 4 °C. The pellet was used for protein extraction as described below. The supernatant was thoroughly mixed with 300 μl xylene, then left shaking at 14000 and 4 °C for 10 min and subsequently centrifuged for 3 min at 22000 x g and 4 °C. The organic phase was discarded and 300 μl of the polar phase were dried in a speed vac for >3 h. For storage, the tube was filled with argon gas and kept at −80 °C until further processing. Directly before running the sample in the GC-MS, the dried aliquot was derivatized as

follows: 20 μl of methoxyaminhydrochlorid (40 mg/ml) in pyridine was added and left shaking for 1.5 h at 30 °C. Subsequently, 180 μl of a mix of N-methyl-N-(trimethylsilyl)trifluoroacetamide (MSTFA) with 20 μl/ml fatty acid methyl ester mix (FAMEs; 0.8 mg/ml of each methyl octanoate, methyl nonanoate, methyl decanoate, methyl laurate, methyl myristate, and methyl palmitate, and 0.4 mg/ml of each methyl stearate, methyl eicosanoate, methyl docosanoate, methyl tetracosanoate, methyl hexacosanoate, methyl octacosanoate, and methyl melissate, in chloroform) was added and left shaking for 30 min at 37 °C. The sample was transferred in a GC-MS vial and analysis was performed on an Agilent 7890 A GC system coupled to a Pegasus HT high throughput TOF/MS (LECO). 1 μl of sample was injected at 230 °C in splitless mode with helium carrier gas flow set to 2 ml/min. Chromatography was performed in 30 meters MDN-35 capillary column, with the following temperature program: isothermal for 2 min at 80 °C, followed by a 15 °C per min ramp to 330 °C, and isothermal for 6 min at 330 °C. Transfer line and ion source temperatures were set to 250 °C. Recorded mass range was set from m/z 70 to m/z 600 at 20 scans per second. Remaining monitored chromatography time was proceeded by a 170-s solvent delay with filaments turned off. Filament bias current was set to −70 V, and detector voltage to -1700–1850 V. Data processing was performed with the Xcalibur™ software (Thermo Fisher) version 4.4.16.14. Metabolite annotation was performed by matching retention indexes based on FAMEs and mass spectra against the Golm Metabolome Database[72]. QC samples corresponding to technical replicates of a pooled mixture of several representative biological samples were included in every batch to correct for systematical error using the SERRF algorithm[73].

## Western Blot analysis

Proteins were extracted from pellets (one per genotype and light condition) obtained from metabolite extraction via adding protein extraction buffer (200 mM Tris, pH 6.8, 8% SDS (w/v), 40% glycerol (v/v) and 200 mM DTT) and heating for 10 min at 65 °C. The proteins were separated according to their molecular weight on SDS-PAGE and blotted on nitrocellulose membrane. Membranes were stained with Ponceau Red (0.1% (w/v) Ponceau S in 5% (v/v) acetic acid) and quantification of the 55 kDa band was used for loading correction. Specific proteins were hybridized with one antibody detecting the four isoforms of ascorbate peroxidase localized in the stroma, thylakoid, peroxisome and cytosol (Agrisera, AS08368; diluted according to manufacturer's instructions) or customized antibodies against HPR1[36] and GGT1/2. The latter was produced in rabbits by Pineda Antikörper Service (Germany) against the peptide sequence CAEEEMPEIMDSFKKFNDE (present in both, GGT1 and GGT2). Signal intensities

were quantified with the GeneTools analysis software relative to WT before the shift and normalized on the Ponceau Red signal from RbcL of each membrane.

## Gas exchange measurements

Gas exchange was measured in 2 s intervals on single leaves from distinct plants using the GFS-3000 open gas exchange system with the LED array unit 3056-FL (with 90% red and 10% blue light) and the optical oxygen sensor 3085-O2 (Walz GmbH, Effeltrich, Germany; Flow: 600 $\mu mol\ s^{-1}$, $CO_2$ concentration: 400 $\mu mol$, relative humidity: 60%, cuvette temperature: 20 °C) on the youngest fully developed leaf in a custom made $3 \times 1$ cm cuvette under either ambient air (21% $O_2$) or low oxygen (2% $O_2$). The $O_2$-dependent effects of $CO_2$ and $H_2O$ absorption were accounted for in measurements of these gasses through an $O_2$ correction based on the concentration measured using the Optical Oxygen sensor 3085-O2. Data was normalized on leaf area, determined from photos via ImageJ.

For measurements under fluctuating light, plants were grown under FL (70 $\mu mol$ photons $m^{-2}\ s^{-1}$, 1 min 700 $\mu mol$ photons $m^{-2}$ $s^{-1}$) for a minimum of 45 d (WT and *ggt2*) or 55 d (*ggt1* and *hpr1* lines), taken directly out of the light and exposed to 45 min light fluctuations (as growth light), then 15 min of 200 $\mu mol$ photons $m^{-2}\ s^{-1}$, and subsequently 30 min darkness. For measurements under control and low light, plants were grown for 39 d (WT and *ggt2*) or at least 50 d (*ggt1* and *hpr1* lines) under FL, allowing the mutants to accumulate sufficient leaf area for the measurement and then shifted to CL or LL. Measurements were conducted over 5 d starting 5 d after the shift.

Net $CO_2$ assimilation rates ($A$) and internal $CO_2$ concentrations ($C_i$) were averaged over the final 10 min of the FL regime and over approximately 1.5 min in steady state of CL and LL. $O_2$-surpressed $CO_2$ assimilation rate ($A_{sup}$) of each plant was calculated as the difference in $CO_2$ assimilation rate under both gas environments.

## Statistical analysis

Most statistical analyses were performed with SigmaPlot version 14.5. For data with small sample size (i.e., absolute metabolite concentration, protein and spectroscopic analyses), normal distribution and equal variance was assumed. To determine the effect of growth condition and genotype, two-way ANOVAs and Tukey multiple comparison tests were conducted. Python (3.7.4) was used to determine significant differences between genotypes in dry weight and relative growth rate by two-sided Games-Howell post-hoc multiple comparison (statsmodels package[74]) and Pearson's linear correlations (SciPy stats package[75]). Statistical comparisons of fluxes between each pair of genotypes within CL and FL conditions were performed by two-sided Wilcoxon rank sum tests for each reaction of interest using the *ranksum* function (Matlab R2020b). The obtained p-values were corrected for multiple hypothesis testing using the Benjamini-Hochberg procedure[76]. Further, the reaction fluxes within each genotype between the CL and FL conditions were compared by two-sided Wilcoxon rank sum tests, followed by Benjamini-Hochberg p-value correction. Spearman correlations between predicted and measured net $CO_2$ assimilation rates were determined using the *corr* function (Matlab R2020b).

## Constraint-based modeling

**Model and data preparation.** The Arabidopsis core model (Ara-Core) was used with updated metabolite names and associated InChI-Keys[51,77]. This was done by inspecting the metabolite names for which mapping (via PubChem) failed and updating them to the appropriate name. Moreover, the localization of the HPR2 protein was updated to the cytosol by introducing the corresponding reaction and a transport reaction for hydroxypyruvate from the peroxisome to the cytosol[37].

For TMFA, compartment-specific pH values were set for the cytosol (pH 7.3)[78], chloroplast (pH 8)[78–80], thylakoid lumen (pH 5)[78,81], mitochondrion (pH 8)[82], mitochondrial intermembrane space (pH 7)[82], nucleus (pH 7.2)[78], and peroxisome (pH 8.4)[78]. The metabolite concentrations in each compartment were allowed to vary between $10^{-8}$ and 0.05 M, as performed in other TMFA studies[83–85].

The matTFA toolbox[86] was then used to associate metabolites in the model with standard Gibbs free energies of formation ($\Delta_f G°$), and reactions with standard Gibbs free energy ($\Delta_r G°$). Since the thermodynamic data provided in the matTFA toolbox are associated to ModelSEED[87] identifiers, metabolites in the AraCore model were translated to ModelSEED identifiers via InChI-Keys. In total, 95.1% of the metabolites and 89.7% of the reactions could be associated with $\Delta_f G°$ or $\Delta_r G°$ values, respectively. In the following, the resulting model is referred to as AraCoreTFA.

Concentrations for 30 metabolites in the AraCoreTFA were obtained from contents ($\mathbf{m}\ [g\ gDW^{-1}]$, $gDW$: gram dry weight) given in Supplementary Data 4 and gathered in the vector $\mathbf{x}$ with entries:

$$x_i = m_i \cdot \frac{\alpha}{\beta} \cdot MW_i^{-1}, \forall i \in M, \qquad (1)$$

where $\alpha$ denotes the weight-to-volume conversion factor ($\alpha = 0.87\ gFW\ l^{-1}$, gFW: gram fresh weight)[88]. Here, $\beta$ denotes the conversion factor from fresh weight to dry weight ($\beta = 10\ gFW\ gDW^{-1}$)[89], $MW_i$ stands for the molecular weight of metabolite $i$, and $M$ is the set of the 30 metabolites with measured content in the AraCoreTFA model. The range of allowed concentration for each metabolite were then calculated by

$$x_i - \sigma_i \leq x_i \leq x_i + \sigma_i, \qquad (2)$$

where $\sigma_i$ denotes standard deviation over four replicates.

**Thermodynamics-based flux analysis.** The AraCoreTFA model was used to predict feasible flux distributions using the TMFA[83,90]. This approach considers $\Delta_r G°$, metabolite concentrations, and temperature to obtain a thermodynamically feasible flux distribution. More precisely, if the Gibbs free energy ($\Delta_r G$) of a reaction $j$ is negative, the reaction proceeds in the forward direction; otherwise, the reaction proceeds in the backward direction. The values for $\Delta_r G$ of individual reactions are determined using the following expressions:

$$\Delta_r G_j = \Delta_r G_j° + \sum_{i=1}^{k} n_{ij}\mu_i, \forall j \in R | \Delta_r G° \text{ is known} \qquad (3)$$

$$\mu_i = \Delta_f G_i° + \Delta_{f,\text{err}} G_i° + RT \ln x_i. \qquad (4)$$

In Eq. (3), the Gibbs free energy of a reaction $j$, $\Delta_r G_j$, is determined by adding the sum of chemical potentials $\mu_i$ of its substrates and products, multiplied by the stoichiometric coefficient of metabolite $i$ in reaction $j$ ($n_{ij}$) to $\Delta_r G°$. The values for $n_{i,j}$ are contained in the stoichiometric matrix $\mathbf{S}$ of the AraCoreTFA model. The chemical potential is calculated in Eq. (4) using the $\Delta_f G°$ values of the metabolite and the associated error term $\Delta_{f,err} G°$; $R$ and $T$ denote the universal gas constant and temperature, respectively.

Further, the relation between $\Delta_r G$ and the direction of the reaction is established by the following two inequalities:

$$\Delta_r G_j - K + Ky_j < 0, \forall j \in R | \Delta_r G° \text{ is known} \qquad (5)$$

$$v_j - Ky_j < 0. \qquad (6)$$

Here, $y_j$ is a binary variable that either has a value of one if the reaction proceeds in the forward direction (i.e., $\Delta_r G_j < 0$), or zero otherwise ($\Delta_r G_j \geq 0$). Further, $K$ is a large number to guarantee that the expression in Eq. (5) is satisfied in the case of $y_j = 0$. Equation (6) provides the coupling of the binary variable to the flux of the reaction.

The full TMFA optimization problem that is solved in the matTFA toolbox is given by

$$\max v_{bio} \tag{7}$$

s.t. (subject to)

$$Sv = 0 \tag{8}$$

$$\begin{aligned} v^{min} \leq v \leq v^{max} \\ v - Ky < 0 \end{aligned} \tag{9}$$

$$\begin{aligned} \Delta_r G - K + Ky < 0 \\ \Delta_r G = \Delta_r G^\circ + S^T \mu. \\ 10^{-8}\,M \leq x \leq 0.05\,M \\ y \in \{0,1\}. \end{aligned} \tag{10}$$

As in flux balance analysis[91], the production of biomass precursors, i.e., flux through the biomass reaction, $v_{bio}$, is maximized (see Eq. (7)). We selected the light-limited biomass formulation in the AraCoreTFA model (*Bio_opt*) for all predictions. The flux through this reaction has the unit $h^{-1}$ and therefore provides the predicted relative growth rate ($v_{bio}$). Equation (8) captures the assumption of metabolic steady state, where **S** is the stoichiometric matrix as described above. The vector **v** contains the flux distribution, i.e., predicted flux through each reaction in the model. We chose the maximum allowed flux value ($v^{max}$) for all reactions to be $1000\,mmol\,gDW^{-1}h^{-1}$ because the flux through the light reactions in the AraCore model is typically higher than $100\,mmol\,gDW^{-1}h^{-1}$ (as used in other TMFA studies). We note that the predicted fluxes through reactions other than light reactions are well below $100\,mmol\,gDW^{-1}h^{-1}$. The minimum flux for all reactions was set to $v^{min} = -1000\,mmol\,gDW^{-1}h^{-1}$. The lower and upper limits for metabolite concentrations were $10^{-8}\,M$ and $0.05\,M$ as described above.

Upon testing the feasibility of the AraCoreTFA model (i.e., testing if $v_{bio}$ can carry at least 50% of the flux that has been predicted by flux balance analysis with the AraCore model), it was found that error for the $\Delta_r G^\circ$ value of the DAP aminotransferase ($\Delta_r G^\circ = 23.2\,kcal\,mol^{-1}$, EC 2.6.1.83) had to be relaxed from $1.4\,kcal\,mol^{-1}$ to $5.4\,kcal\,mol^{-1}$.

**Flux variability analysis and sampling.** The space of alternative solutions (i.e., flux distributions) to the TMFA problem stated above that are associated with the same optimal value for $v_{bio}$ was explored by sampling 1000 random distributions. To this end, first the feasible ranges (interval between minimum and maximum values that a reaction $j$ can take while satisfying the optimal objective value) were determined by thermodynamic-based flux variability analysis (TVA). This was achieved by changing the objective in the TMFA problem, described above, to

$$\min / \max v_j \tag{11}$$

with an additional constraint on the objective value (here $v_{bio}$):

$$v_{bio} = v_{bio}^{opt}. \tag{12}$$

Flux distributions were sampled by projecting a random vector ($v^*$) of flux values within the feasible ranges onto the solution space given by the TMFA problem. Hence, the constraints to the flux

sampling problem are identical to the TVA problem, but the objective is changed to finding the flux distribution with the smallest distance to $v^*$, by minimizing the first norm:

$$\min |v^* - v|. \tag{13}$$

**Modeling of flux distributions in photorespiratory mutants.** The modeling framework detailed above was used to predict flux distributions of the Col-0 wild type and the four photorespiratory mutants *ggt1-1*, *ggt1-2*, *hpr1-1*, and *hpr1-2* in both constant and fluctuating light. To arrive at mutant-specific models, reactions associated with either the GGT1 gene (*GGAT_p*, *AlaTA_p*) or the HPR1 gene (*GCEADH_p*) were blocked. The acclimation of the *hpr1* mutant from high to low $CO_2$ was previously modeled by integrating relative metabolomics data[92]. Further, the model was adjusted to each of the two light conditions by constraining the photon uptake reaction ($h_{hv}$) and the ratio between the oxygenation and carboxylation reaction of Rubisco ($\phi = v_o v_c^{-1}$, Supplementary Data 8). Moreover, ratios between relative growth rates and net $CO_2$ assimilation rates ($A_{net}$) were constrained between the wild-type models in CL and FL condition, as well as between the wild-type and mutant models in the two light conditions. For more details please see the Supplementary methods.

### Reporting summary

Further information on research design is available in the Nature Portfolio Reporting Summary linked to this article.

## Data availability

Source data for Figs. 1–5 and Supplementary Figs. 3–7 & 7–14 are provided with the paper. All raw chromatograms for metabolomics analysis are publicly available as a MassIVE dataset MSV000091798 under the https://doi.org/10.25345/C5377651Q [https://massive.ucsd.edu/ProteoSAFe/dataset.jsp?task=f103606327b84c0b84533de4d59592fe]. Source data are provided with this paper.

## Code availability

All custom code and data generated for the constraint-based analysis is available at https://github.com/pwendering/model-prm. Updates introduced to functionalities of the matTFA toolbox can be found at the GitHub fork https://github.com/pwendering/matTFA.

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

## Acknowledgements

We thank Elias Kaiser and Igor Flórez-Sarasa for valuable discussions and Hermann Bauwe and Stefan Timm for kindly supplying *PGLP*, *PsL* and *gldt1-1* seeds and HPR1 antibody. We acknowledge help with plant harvesting by Urszula Luzarowska, Jakob Keller, Sandrine Kappel, Kübra Korkmaz and Michal Uflewski and help with plant cultivation by the MPI-MP greenteam. TvB, PW, MAS, ARF, ZN and UA were supported by the Max Planck society. UA was funded by the Deutsche Forschungsgemeinschaft (DFG, German Research Foundation) under

Germany´s Excellence Strategy – EXC-2048/1 – project ID 390686111". Figure 6 was created using BioRender.com.

## Author contributions

U.A. and T.v.B. designed the study. T.v.B. performed most experimental work with support from J.R. and L.S. P.W. and Z.N. designed the constraint-based modeling procedures and analyzed the findings. P.W. implemented and performed constraint-based modeling. L.P.S. conducted the data processing of MS data. M.A.S. and B.W. helped with the gas exchange experiments. E.H. provided input in the early stages and B.W. at the later stages of the study. T.v.B., P.W., L.P.S., M.A.S., A.F., Z.N., and U.A. interpreted data. T.v.B. and U.A. wrote the manuscript with contributions and input from all authors.

## Funding

## Competing interests

The authors declare no competing interests.
