## [Peer Review File · Nature Communications]

Growth in fluctuating light buffers plants against photorespiratory perturbationsREVIEWER COMMENTS

Reviewer #1 (Remarks to the Author):

Bismarck et al

The authors report a significant study of the impact of various genetic perturbations to photorespiration on growth, photosynthesis and photorespiration under contrasting light regimes. Rather surprisingly, they find that fluctuating light does not lead to stronger perturbations to phenotype, but rather the opposite. This is in contrast to other sections of the photosynthesis literature that are increasingly being populated by work showing that fluctuating light leads to clearer changes in mutant or transgenic lines. The conclusions from this current study will have broad reach.

Interestingly, the data point to the notion that weak mutants in photorespiration cope better in fluctuating light because the changes in photosynthetic flux mean that phosphoglycolate does not accumulate to the same levels as in constant light, and so required fluxes through photorespiration are lower. Data supporting this general contention are provided from chlorophyll fluorescence, metabolite analysis and running a flux model. The results seem to end on the intriguing possibility that an increase in H₂O₂ production in the chloroplast is key to this. One could argue that this could be tested with either a ROS sensitive dye, or a reporter, but I feel that the general finding that fluctuating light has reduced photorespiratory load is so surprising that it should be published without this. The notion that fluctuating light allows metabolism to be re-routed constantly, and that this has beneficial effects is novel in my view.

Introduction: sets the scene nicely.

Results: Transcriptional co-regulation is misleading. The authors did not measure transcription per se. Rather they have analysed transcript abundance that is determined by RNA half life, degradation as well as rate of production. Please rephrase.

Comparison with the classic work from Stefan Jansson on when NPQ is useful ie under fluctuating light conditions, and other studies highlighting how processes can be engineered

for use in fluctuating conditions.

I think the authors could explicitly comment on whether they think this sensitivity to light regime is likely to be phylogenetically variable. And to include some text on what this means for the various work that has and is implementing alternative pathways to increased the efficiency of photorespiration.

Reviewer #2 (Remarks to the Author):

The present paper tried to elucidate the importance of photorespiration in fluctuating light conditions. The many experiments were done. However, the reviewer cannot understand the logic of the present story.

1. What is the control condition for the normal growth?
2. How do authors set the time duration of the present experiments for the fluctuating light? It was un-natural. What was the physiological meanings? Why did the authors use the fluctuating light?
3. The authors cited the reference (26-28). They emphasized the importance of photorespiration to protect photosystem I, and led the readers to the physiological meanings of the existence of photorespiration in the photosynthesis organisms. The reviewer cannot understand what happens in their references. If ATP were much used in photorespiration, compared to the net CO₂ assimilation, the photosynthesis control to down-regulate the plastoquinol oxidation activity by cytochrome b₆/f-complex was suppressed to lead to the much reduction of P700 in photosystem I. These situations stimulated the reduction of the Fe/S-clusters, the electron carriers at the acceptor-side of photosystem I, resulting in the enhancement of reactive oxygen species (ROS) production. Therefore, the reviewer cannot understand the author`s story. How does photorespiration protect photosystem I?
4. Unless the answers to the above great issues, the reviewer cannot understand why the authors used the mutants in the manuscript?
5. What is the definition of the growth of plants in the present research? If the intensity of light under control was different from the present intensity, the results should differ from

the present results. Arabidopsis can grow even at the twice intensity of light used in the present intensity.

6. The present research shows the non-requirement of photorespiration for the growth under the fluctuating light conditions the authors set. From the results in Figure 2, C_i was high under both low light and fluctuating light conditions. These situations stimulate the net CO_2 assimilation to support their growth.

7. Line 312: ..., despite both conditions providing the same average light intensity over the day. => What does this mean? This is not the reason for the control condition of CL against the fluctuating light.

From the above matters, the reviewer do not agree with the statement of the authors (Our results uncover that dynamic light environments in the abstract), and what is the worse, the authors did not consider the experiments for the dynamic light environments.

Reviewer #3 (Remarks to the Author):

The article studies the function of the phosphorespiratory pathway (PR) in plants under fluctuating and non-fluctuating light conditions, and the effects of altered levels of PR enzymes. The authors found that plants with decreased PR enzyme levels are less affected by light fluctuations, and seek to explain this unintuitive finding by a combination of measurements and modelling. The article is very well written, results are clearly explained and the findings are supported by a good amount of data.

The authors apply thermodynamic metabolic flux analysis (TMFA) to a core model of Arabidopsis metabolism, which is a previously known technique. The implementation looks well conducted and I find it noteworthy that the authors took care to explain the method again in good level of detail, which will certainly help unfamiliar readers. The source code has been shared on GitHub and metabolomics data has been deposited in a public database, which is important too for reproducibility. I have a few questions about details of the method as follows:

1. Biomass production has been used as the objective for metabolic flux calculations. It has

been documented that this is generally a good objective for microorganisms, but less so for more complex organisms such as mammals and plants. Is this the best objective to study plants under conditions of fluctuating light, where other cellular objectives such as saving resources may be more important than growth? Even if it isn't possible to repeat all computations with different objective functions, I think this point should be addressed in the discussion.

2. On line 503 the authors mention updated metabolite names, how and why were they updated?

3. On line 588, how does this flux sampling method relate to other known methods? Have the authors tested its homogeneity and convergence?

REVIEWER COMMENTS

Please note that references specific for this response letter are referred to by the style “Author (Date)” and can be found at the end of this document. References from repeats of the manuscript in this letter (in quotation marks) are referred to by numbers in subscript and can be found only in the manuscript.

Reviewer #1 (Remarks to the Author):

Bismarck et al

The authors report a significant study of the impact of various genetic perturbations to photorespiration on growth, photosynthesis and photorespiration under contrasting light regimes. Rather surprisingly, they find that fluctuating light does not lead to stronger perturbations to phenotype, but rather the opposite. This is in contrast to other sections of the photosynthesis literature that are increasingly being populated by work showing that fluctuating light leads to clearer changes in mutant or transgenic lines. The conclusions from this current study will have broad reach.

Interestingly, the data point to the notion that weak mutants in photorespiration cope better in fluctuating light because the changes in photosynthetic flux mean that phosphoglycolate does not accumulate to the same levels as in constant light, and so required fluxes through photorespiration are lower. Data supporting this general contention are provided from chlorophyll fluorescence, metabolite analysis and running a flux model. The results seem to end on the intriguing possibility that an increase in H₂O₂ production in the chloroplast is key to this. One could argue that this could be tested with either a ROS sensitive dye, or a reporter, but I feel that the general finding that fluctuating light has reduced photorespiratory load is so surprising that it should be published without this. The notion that fluctuating light allows metabolism to be re-routed constantly, and that this has beneficial effects is novel in my view.

We would like to thank Reviewer 1 for the positive evaluation of the manuscript.

1) Introduction: sets the scene nicely.

We made some adjustments to the introduction as requested by Reviewer 1 and 2.

2) Results: Transcriptional co-regulation is misleading. The authors did not measure transcription per se. Rather they have analysed transcript abundance that is determined by RNA half life, degradation as well as rate of production. Please rephrase.

Thanks for noting the misleading expression. In response, we exchanged “co-regulation” with “co-expression” in the heading of the paragraph (ln 88) and “transcriptionally regulated” with “co-regulated at the transcript level” (ln 91) within the paragraph. We hope that these adjustments clarify sufficiently that we refer to co-regulation on transcript level instead of transcriptional activity.

3) Comparison with the classic work from Stefan Jansson on when NPQ is useful ie under fluctuating light conditions, and other studies highlighting how processes can be engineered for use in fluctuating conditions.

In response, we restructured the introduction to focus on fluctuating light as a challenge for plants and potentially limiting responses of regulatory mechanisms. Thereby, we introduced PR as a regulatory mechanism that was proposed to counteract some of the kinetic inefficiencies of the regulatory processes. The new paragraph (lns 56-70) now reads:

“Under natural sunlight conditions, plants are often exposed to strong and rapid fluctuations in light intensity²¹. These fluctuations induce regulatory mechanisms, which are important for plant fitness in the field²². The responses of some of these regulatory mechanisms are much slower than changes in light intensity and it was proposed that kinetic inefficiencies may limit crop yield²³⁻²⁷. Directly following a transition from low to high light, delays in the induction of stomatal conductance and the CBB cycle lead to a transient decrease in CO₂ supply to Rubisco^{21,28} and overproduction of ATP and NADPH that cannot be used by the CBB cycle. It was hypothesized that, in fluctuating light, PR plays a role for counteracting the kinetic inefficiencies of these two processes, by: (i) providing increased rates of 2PG metabolization needed because of higher Rubisco oxygenation per carboxylation rates²⁹ and (ii) simultaneously serving a sink for the transient spikes of metabolic energy produced by the light reactions, thereby avoiding photo-oxidative damage to the photosynthetic machinery²⁹⁻³¹. For *Arabidopsis thaliana* (Arabidopsis) grown in fluctuating light it was shown that they accumulated higher transcript and protein levels of some PR enzymes, suggesting that PR metabolism is upregulated^{32,33}.”

I think the authors could explicitly comment on whether they think this sensitivity to light regime is likely to be phylogenetically variable. And to include some text on what this means for the various work that has and is implementing alternative pathways to increased the efficiency of photorespiration.

This is an interesting consideration that we addressed with changes to the discussion: Firstly, we now propose to identify the enzymes and signal(s) responsible for chloroplast localized glycolate decarboxylation. This may allow implementation of this endogenous PR pathway to improve (C₃) crops. For such strategy to be useful, we require additional information on plant species- and fluctuating-light specific differences in the engagement of this PR bypass (which may additionally reveal the molecular nature of the involved enzymes and underlying signals). The discussion now reads (lns 371-379):

“Synthetic PR bypasses within the chloroplast that introduced glycolate converting enzymes into this organelle were shown, under some conditions, to increase biomass of field-grown tobacco¹². Uncovering the enzymes responsible for the endogenous bypass revealed in this study and the signal(s) that control its activity may provide avenues to improved crop performance. As PR metabolism is strongly conserved amongst higher plants (reviewed in ref. ⁵⁸), such strategies may be applicable to most C₃ crops. With this prospective, it still remains to be determined whether fluctuating light causes similar re-routing of PR and other metabolic pathways in crops and to which extent natural dynamic light conditions differ in their effects.”

Reviewer #2 (Remarks to the Author):

The present paper tried to elucidate the importance of photorespiration in fluctuating light conditions. The many experiments were done. However, the reviewer cannot understand the logic of the present story.

We hope that we could address your concerns sufficiently by providing further information on the rationale and methods of the study.

1. What is the control condition for the normal growth?
2. How do authors set the time duration of the present experiments for the fluctuating light? It was un-natural. What was the physiological meanings? Why did the authors use the fluctuating light?

Answers to 1 & 2:

We removed the “natural” from the abstract (ln 21) to avoid confusion.

Highly fluctuating light conditions were used as described. Light regimes, which subject plants to repetitive 10-fold changes in light intensity, in comparison to a control condition, which provides the same average light intensity over the day, have been used before by us as well as multiple other groups (e.g., Niedermaier et al. 2020; Schneider et al. 2019; Suorsa et al. 2012; Garcia-Molina and Leister 2020; Dukic et al. 2019; Fristedt et al. 2017; von Bismarck et al. 2023; Armbruster et al. 2016; Uflewski et al. 2021). The use of standardized fluctuating light regime is important for comparison of results and a comprehensive understanding of plant physiology under the given condition.

For further explanation within the manuscript, we included a paragraph addressing the relevance of fluctuating light in the introduction (please see point 3).

The two constant growth light conditions CL and LL were chosen as controls for the fluctuating light regime because (i) CL exposed the plants to the same average growth light intensity, but served in a constant manner, and (ii) LL resulted in the same average CO₂ assimilation rate. To clarify this in the text, we extended the explanation of the specific growth light conditions (lns 102-114):

“To investigate the role of PR in dynamic light, we grew Arabidopsis plants with different modifications of PR metabolism under one strongly fluctuating light and two non-fluctuating light regimes. Besides wild type (WT), a newly characterized *ggt2* T-DNA insertion line (Supplementary Fig. S1d), the previously published “mild” PR mutants *ggt1*, *hpr1* and *gldt1-1*^{36,39-41}, and two lines with reported higher PR flux (overexpression of *PGLP* or the L protein of GCS, i.e., *PsL*^{3,46}) were used (Fig. 1a). The growth light was provided at a 12 h photoperiod with 144 repetitions of 10-fold changes in light intensity (4 min 70 μmol photons m⁻² s⁻¹ and 1 min 700 μmol photons m⁻² s⁻¹) for the strongly fluctuating light (FL) condition, 200 μmol photons m⁻² s⁻¹ for the control light (CL) condition with the same average daily photon flux density as FL, and 90 μmol photons m⁻² s⁻¹ for a low light (LL) control condition intended to account for similar rates of CO₂ fixation as in FL.”

3. The authors cited the reference (26-28). They emphasized the importance of photorespiration to protect photosystem I, and led the readers to the physiological meanings of the existence of photorespiration in the photosynthesis organisms. The reviewer cannot understand what happens in their references. If ATP were much used in photorespiration, compared to the net CO₂ assimilation, the photosynthesis control to down-regulate the plastoquinol oxidation activity by cytochrome b6/f-complex was suppressed to lead to the much reduction of P700 in photosystem I. These situations

stimulated the reduction of the Fe/S-clusters, the electron carriers at the acceptor-side of photosystem I, resulting in the enhancement of reactive oxygen species (ROS) production. Therefore, the reviewer cannot understand the author's story. How does photorespiration protect photosystem I?

We understand the reviewer may refer to the differential energy requirements between the CBB cycle and PR (higher ATP per NADPH requirement for PR). In response to high PR rates, low ATP levels would lead to a high ATP synthase conductivity for protons, decreased proton concentration in the lumen (higher lumen pH) and thus low photosynthetic control at the Cyt *b₆f* complex, thereby further enforcing PSI acceptor side limitation. This is a valid remark from a photosynthesis point of view! In this manuscript, we state previous hypotheses by other studies that proposed PR to occur at higher rates in fluctuating light (because of lower C_i concentrations) and thereby serve an energy sink function. We show that Rubisco oxygenation reaction is highest for our CL condition and comparable between FL and LL and thus disprove that PR requirements are increased under FL, implicating that also no extra transient sink is needed.

4. Unless the answers to the above great issues, the reviewer cannot understand why the authors used the mutants in the manuscript?

We hope that the reviewer's concerns are sufficiently addressed by the answers to comments 1-3. The mutants were used to study the effect of differences in photorespiratory metabolism for plant growth under fluctuating and non-fluctuating light conditions. We then focused our study on understanding, why mutants with lower photorespiratory flux than WT grow well under highly fluctuating light conditions.

5. What is the definition of the growth of plants in the present research?

For the growth analysis, we determined dry weight and from this calculated the relative growth rate per day (see Ln 115), as explained in the Material & Methods section (Lns 456-457 and Lns 603-605) and in Supplemental Figures S3 and S4. To clarify, we included the definition of relative growth rate, i.e., the rate of accumulation of new dry mass per unit of existing dry mass per day, into the legend of Fig. 1 and Supplemental Fig. S4.

If the intensity of light under control was different from the present intensity, the results should differ from the present results. Arabidopsis can grow even at the twice intensity of light used in the present intensity.

CO₂ assimilation capacity correlates strongly with growth light intensity (von Bismarck et al. 2023), so we expect higher rates of PR if the plants are grown under high light. Thus, photorespiratory mutants should show a stronger relative growth reduction compared with WT with increasing light intensities. What is exciting about our results is that the growth rate of *ggt1* does not solely relate with the rate of CO₂ assimilation/PR, but that FL has a genuinely positive effect likely by suppressing chloroplast H₂O₂ production. We do agree with the reviewer that the results of this study may not be consistent under other fluctuating light conditions, as these can vary strongly in several factors like light intensity, duration of the high light spike, frequency of the fluctuation and so on. Future work is needed to systematically assess these effects. We refer to this in the discussion (Lns 376-379):

“With this prospective, it still remains to be determined whether fluctuating light causes similar re-routing of PR and other metabolic pathways in crops and to which extent natural dynamic light conditions differ in their effects.”

6. The present research shows the non-requirement of photorespiration for the growth under the fluctuating light conditions the authors set. From the results in Figure 2, C_i was high under both low light and fluctuating light conditions. These situations stimulate the net CO₂ assimilation to support their growth.

In general, the reviewer is correct, in that C_i is slightly higher both under fluctuating and low light (approximately 360 ppm) than under constant light (340 ppm). However, these differences in C_i are so minor that effects on the relative rate of photorespiration (oxygenation per carboxylation reaction) are not detectable, as stated in the text (see below). Also, the reviewer might note that despite the same C_i values under LL and FL conditions, only under the FL regime, relative growth of the photorespiratory mutants was restored to wild-type levels, while in LL, this was not the case. Please see lns 152-157:

“The similar A_{sup} and C_i values across genotypes for each condition suggested that relative rates of oxygenation per carboxylation reactions were comparable to WT in both *hpr1* and *ggt1* (Supplementary Table S1). All gas exchange parameters determined for *ggt2* were WT-like. The small increase in C_i from 340 (CL) to 360 ppm (FL) did not efficiently repress photorespiration, as evident by similar A_{sup} between both conditions, and therefore cannot explain the better growth of the mutants under FL relative to CL.”

Additionally, the ability of 20 ppm more CO₂ to stimulate photosynthesis in WT and photorespiratory mutants is small, for example, an increase from 340 to 360 ppm would only increase net assimilation by 1-2% under saturating light as determined from CO₂ response curves measured in WT and *hpr1-1* mutants in past work (Cousins et al. 2011).

7. Line 312: ..., despite both conditions providing the same average light intensity over the day. => What does this mean? This is not the reason for the control condition of CL against the fluctuating light.

To clarify, we changed the sentence to the following (lns 328-329):

“All analyzed genotypes displayed higher rates of photosynthesis under CL compared with FL (Fig. 2a), with both conditions providing the same average light intensity over the day.”

From the above matters, the reviewer do not agree with the statement of the authors (Our results uncover that dynamic light environments in the abstract), and what is the worse, the authors did not consider the experiments for the dynamic light environments.

We hope that our answers to comments above, as well as further clarifications in the manuscript sufficiently address the concerns of the reviewer.

Reviewer #3 (Remarks to the Author):

The article studies the function of the phosphorespiratory pathway (PR) in plants under fluctuating and non-fluctuating light conditions, and the effects of altered levels of PR enzymes. The authors found that plants with decreased PR enzyme levels are less affected by light fluctuations, and seek to explain this unintuitive finding by a combination of measurements and modelling. The article is very well written, results are clearly explained and the findings are supported by a good amount of data.

We would like to thank Reviewer 3 for the positive evaluation of the manuscript.

The authors apply thermodynamic metabolic flux analysis (TMFA) to a core model of Arabidopsis metabolism, which is a previously known technique. The implementation looks well conducted and I find it noteworthy that the authors took care to explain the method again in good level of detail, which will certainly help unfamiliar readers. The source code has been shared on GitHub and metabolomics data has been deposited in a public database, which is important too for reproducibility. I have a few questions about details of the method as follows:

1. Biomass production has been used as the objective for metabolic flux calculations. It has been documented that this is generally a good objective for microorganisms, but less so for more complex organisms such as mammals and plants. Is this the best objective to study plants under conditions of fluctuating light, where other cellular objectives such as saving resources may be more important than growth? Even if it isn't possible to repeat all computations with different objective functions, I think this point should be addressed in the discussion.

We agree with the reviewer that this is an important issue in constraint-based modelling analyses of multicellular organisms and we now address this in the discussion (Ins 399-405). We argue that the accumulation of biomass certainly is an important objective that is optimized by a plant under standard conditions (during the vegetative phase). Since the model that we used is a leaf model, we therefore assume maximization of biomass accumulation, i.e., maximization of the relative growth rate for the wild type under constant light conditions (first optimization problem that is solved). For all subsequent problems (mutants, and fluctuating light condition), we did not optimize the relative growth rate. Instead, we fixed the modeled relative growth rates relative to the growth rate that we predicted from the first optimization problem by using the relative growth rates that we calculated from the measured dry weights.

There could be another objective that is optimized by the plants under fluctuating light, which, if known, could shape the flux distribution when considered in the optimization. However, finding alternative objective(s) for the stress condition exceeds the scope of this study. That said, we would like to stress that rerouting of fluxes due to the stress condition or introduced knock-outs is considered by including the metabolite abundances in the TMFA formulation.

Therefore, we only assume optimization of the relative growth rate for the wild type under constant light and use known ratios of relative growth rates to the mutants and stress condition to constrain the solution space. We do, however, address two additional objectives, which are based on the following assumptions (1) efficient use of resources, i.e., parsimonious enzyme usage (minimization of the sum of fluxes, pFBA) and (2) minimal rerouting of fluxes between the wild type and mutants (minimization of the first norm to the wild type flux vector). The latter objective has been successfully used to model mutants of *E. coli* (e.g., (Segrè, Vitkup, and Church 2002)) and *A. thaliana* (Sajitz-Hermstein et al. 2016).

We added the following paragraph to the discussion to address the use of biomass as the objective for the metabolic flux calculations (Ins 400-406):

“We used a metabolic model of a growing leaf; hence we assume maximization of the RGR for the WT under constant light as the primary objective for TMFA optimization. For all other considered scenarios (i.e., modeling of mutants and FL condition), we did not optimize the RGR, because growth in these cases may not present the primary objective; instead, we fixed the RGRs, which were calculated from the measured dry weights, relative to the WT-RGR of CL. This procedure should account for differences in primary objectives given by the fluctuating light or PR-enzyme deficiencies.”

2. On line 503 the authors mention updated metabolite names, how and why were they updated?

Some of the metabolite names in the model were updated manually to improve the automated mapping of metabolite names to InChI-keys. This was done by inspecting the metabolite names for which mapping (via PubChem) failed and updating them to the appropriate name. We included this in the Material and Method section as follows (Ins 545-546):

“This was done by inspecting the metabolite names for which mapping (via PubChem) failed and updating them to the appropriate name.”

3. On line 588, how does this flux sampling method relate to other known methods? Have the authors tested its homogeneity and convergence?

The reviewer is asking an important question, which was not addressed in the first version of the manuscript due to the presence of binary variables in the optimization problem. We added a new section to the Supplementary Information (please see last section “Effects of alternative flux sampling”) and address this in the discussion (Ins 419-420). We agree with the reviewer that incomplete coverage of the solution space by flux sampling may lead to misleading results with respect to the average fluxes. In the paper that accompanies the matTFA toolbox (Salvy et al. 2018), which we used to construct the TMFA model, the authors fixed the directions of bidirectional reactions to be able to use the artificial centering hit-and-run (ACHR) sampler implemented as part of the COBRA toolbox. Since this procedure decreases the feasible space to be sampled in a biased way, we developed the sampling procedure described in our study to directly use the TMFA problem for sampling. While this approach allows a fast flux sampling, we are aware that it may not guarantee uniform sampling of the feasible space.

To resolve this issue, for a comparison, we applied the gapsplit sampler (Keaty and Jensen 2020), which can deal with mixed-integer linear problems. One representative optimization problem (Col-0, CL) was exported and used as input for the gapsplit sampler with default settings. All net flux variables (the same variables, for which random vectors we projected onto the solution space) were selected as primary targets whose distributions in the feasible space are explored by gapsplit. To ensure feasibility of the second norm minimization for the secondary targets, we first identified and corrected an error in the gapsplit code and set an upper limit for the minimization weights of 10,000. The performances of the two sampling strategies, gapsplit and our projection-based method, were compared by using the coverage, defined as follows (Keaty and Jensen 2020; Binns et al. 2015):

$$\text{coverage} = 1 - \text{mean}(\text{relative max gap}(x_i)) .$$

The term $\max\text{gap}(x_i)$ denotes the maximum difference between any two sampled values including the limits of the feasible ranges of a variable x_i . The relative max gap is then calculated by scaling $\max\text{gap}(x_i)$ by the feasible range of x_i ($\max(x_i) - \min(x_i)$).

We observed that our sampling strategy (“projection”) only reached a maximum coverage of 28.25% with 985 samples, while, with the same number of samples, the gapsplit sampler reached a coverage of 86.4% (Fig. S16a). We further note that the “projection” sampling uses first norm minimization to minimize the distance between the random flux vector and the feasible space, while the gapsplit sampler uses second norm minimization to minimize the distance of average gaps of secondary targets to the feasible space (with fixed primary target). The use of the second norm results in many small differences, while using the first norm results in fewer, large differences. Hence, the use of the second norm in gapsplit increases the coverage by adding many small differences, but these may not necessarily affect the average flux values that we compared in this study (in different modeling scenarios).

To investigate whether or not the difference in sampling coverage yields significant changes with respect to average fluxes, we compared the average values of all net flux variables (Fig. S16b). We found that the average flux values obtained from the two sampling approaches correlated perfectly, with a Pearson correlation of $\rho = 0.9999$. Notably, some fluxes only showed very low average flux values ($< 10^{-10} \text{ mmol gDW}^{-1} \text{ h}^{-1}$) when using the “projection” sampling, which is a result of incomplete coverage, already discussed above. These reactions are also mainly responsible for the disagreement in sign of the respective averages in flux values between the two approaches, which otherwise show high agreement (88.4%) of the flux averages across the reactions in the model. Further, the standard deviations (Fig. S16c) and ranges (Fig. S16d), difference between maximum and minimum sampled flux) agree with high Pearson correlations of 0.96 and 0.99, respectively. From Fig. S16c-d, it is then obvious that the disagreements in sign result from the very narrow ranges (and small standard deviations) for some of the reactions, which cause numerical fluctuations around average fluxes that are essentially equal to zero – that can be readily neglected.

Five reactions with average fluxes below $10^{-10} \text{ mmol gDW}^{-1} \text{ h}^{-1}$ from the projection sampling are presented in Figure 5; however, the square root of the average flux obtained using gapsplit does not exceed $0.04 \text{ mmol gDW}^{-1} \text{ h}^{-1}$, which is very low compared to the remaining fluxes that are shown. These include: the chloroplastic HPR3 reaction showed $\sqrt{v} = 0.04 \text{ mmol gDW}^{-1} \text{ h}^{-1}$, which is in line with the other very low-magnitude fluxes shown in the heatmap. Three reactions are involved in the conversion of 2-oxoglutarate to succinate and had $\sqrt{v} = 0.005 \text{ mmol gDW}^{-1} \text{ h}^{-1}$. The fifth reaction is the mitochondrial NADH-dependent isocitrate dehydrogenase, which had $\sqrt{v} = 0.0026 \text{ mmol gDW}^{-1} \text{ h}^{-1}$. While these low average fluxes would have caused dark blue coloring in the heatmaps in Figure 5, they would still have been considered marginally active.

From these results, we conclude that while the use of a different sampling strategy (i.e., gapsplit) provided a better coverage of the feasible flux space, the used projection sampling approach does not affect the presented findings with respect to the average fluxes as well as standard deviations per reaction. Therefore, the findings presented in the first version of the manuscript are not affected by the sampling approach used and remain valid.

We added the following sentence to the discussion (Ins 420-421):

“A second flux sampling approach⁶⁴, providing a wider coverage of the flux space, showed robustness of the conclusions (Supplementary methods, Supplementary Fig. S16).”

Fig. S16. Comparison of flux sampling results with gapsplit for one representative optimization problem.

The first optimization problem (i.e., Col-0, CL) was used as input for the gapsplit sampler (Keaty and Jensen 2020) with default settings. The same variables for which random vectors were projected onto the feasible space were used as primary targets in gapsplit (i.e., net flux variables). **(a)** comparison of the “projection” sampling approach used in this study and the gapsplit sampler sampling by coverage defined by $1 - \text{average relative maximum gap}$ (considering only the net flux variables, number of samples: 985). Panels **(b-d)** show absolute average fluxes **(b)**, standard deviations **(c)**, and ranges **(d)** (differences between minimum and maximum flux) over all samples. Black color and red color indicate agreement in positive and negative sign, respectively (81.7% and 6.8%, total 88.4%), and blue color indicates disagreement in signs (11.6%) (panels (b-d)). The gray line in panels (b-d) depicts perfect agreement.

References

- Armbruster, U., Leonelli, L., Correa Galvis, V., Strand, D., Quinn, E. H., Jonikas, M. C., and Niyogi, K. K. 2016. 'Regulation and Levels of the Thylakoid K⁺/H⁺ Antiporter KEA3 Shape the Dynamic Response of Photosynthesis in Fluctuating Light', *Plant Cell Physiol*, 57: 1557-67.
- Binns, M, de Aauri, P., Vlysidis, A., Cascante, M., and Theodoropoulos, C. 2015. 'Sampling with pooling-based flux balance analysis: optimal versus sub-optimal flux space analysis of *Actinobacillus succinogenes*', *BMC Bioinformatics*, 16: 49.
- Cousins, A. B., Walker, B. J., Pracharoenwattana, I., Smith, S. M., and Badger, M. R. 2011. 'Peroxisomal hydroxypyruvate reductase is not essential for photorespiration in *Arabidopsis* but its absence causes an increase in the stoichiometry of photorespiratory CO₂ release', *Photosynth Res*, 108: 91-100.
- Dukic, E., Herdean, A., Cheregi, O., Sharma, A., Nziengui, H., Dmitruk, D., Solymosi, K., Pribil, M., and Spetea, C. 2019. 'K⁺ and Cl⁻ channels/transporters independently fine-tune photosynthesis in plants', *Scientific Reports*, 9: 8639.
- Fristedt, R., Trotta, A., Suorsa, M., Nilsson, A. K., Croce, R., Aro, E., and Lundin, B. 2017. 'PSB33 sustains photosystem II D1 protein under fluctuating light conditions', *Journal of Experimental Botany*, 68: 4281-93.
- Garcia-Molina, A., and Leister, D. 2020. 'Accelerated relaxation of photoprotection impairs biomass accumulation in *Arabidopsis*', *Nature Plants*, 6: 9-12.
- Keaty, T. C., and Jensen, P. A. 2020. 'Gapsplit: efficient random sampling for non-convex constraint-based models', *Bioinformatics*, 36: 2623-25.
- Niedermaier, S., Schneider, T., Bahl, M., Matsubara, S., and Huesgen, P. F. 2020. 'Photoprotective Acclimation of the *Arabidopsis thaliana* Leaf Proteome to Fluctuating Light', *Front Genet*, 11.
- Sajitz-Hermstein, M., Töpfer, N., Kleessen, S., Fernie, A. R., and Nikoloski, Z. 2016. 'iReMet-flux: constraint-based approach for integrating relative metabolite levels into a stoichiometric metabolic models', *Bioinformatics*, 32: i755-i62.
- Salvy, P., Fengos, G., Ataman, M., Pathier, T., Soh, K. C., and Hatzimanikatis, V. 2018. 'pyTFA and matTFA: a Python package and a Matlab toolbox for Thermodynamics-based Flux Analysis', *Bioinformatics*, 35: 167-69.
- Schneider, T., Bolger, A., Zeier, J., Preiskowski, S., Benes, V., Trenkamp, S., Usadel, B., Farré, E. M., and Matsubara, S.. 2019. 'Fluctuating Light Interacts with Time of Day and Leaf Development Stage to Reprogram Gene Expression', *Plant Physiol*, 179: 1632-57.
- Segrè, D., Vitkup, D., and Church, G.M. 2002. 'Analysis of optimality in natural and perturbed metabolic networks', *Proceedings of the National Academy of Sciences*, 99: 15112-17.
- Suorsa, M., Järvi, S., Grieco, M., Nurmi, M., Pietrzykowska, M., Rantala, M., Kangasjärvi, S., Paakkanen, V., Tikkanen, M., Jansson, S., and Aro, E. 2012. 'PROTON GRADIENT REGULATION5 Is Essential for Proper Acclimation of *Arabidopsis* Photosystem I to Naturally and Artificially Fluctuating Light Conditions', *The Plant Cell*, 24: 2934-48.
- Uflewski, M., Mielke, S., Correa Galvis, V., von Bismarck, T., Chen, X, Tietz, E., Ruß, J., Luzarowski, M., Sokolowska, E., Skirycz, A., Eirich, J., Finkemeier, I., Schöttler, M. A., and Armbruster, U. 2021. 'Functional characterization of proton antiport regulation in the thylakoid membrane', *Plant Physiol*, 187: 2209-29.
- von Bismarck, T., Korkmaz, K., Ruß, J., Skurk, K., Kaiser, E., Correa Galvis, V., Cruz, J. A., Strand, D. D., Köhl, K., Eirich, J., Finkemeier, I., Jahns, P., Kramer, D. M., and Armbruster, U. 2023. 'Light acclimation interacts with thylakoid ion transport to govern the dynamics of photosynthesis in *Arabidopsis*', *New Phytol*, 237: 160-76.

REVIEWERS' COMMENTS

Reviewer #2 (Remarks to the Author):

Thank you so much for your description to my reviewing, and your revision of the manuscript. Although I can not understand the explanation of the logic of experiments using FL, at this stage any more results and facts would not be required.

As you say that you find the different behaviors of mutants from WT, it is fact and OK. You say FL conditions are usual growth conditions under the natural fields. In future, I would like to see what happens when WT and your mutants grow under natural light conditions (GMO cabinet).

Reviewer #3 (Remarks to the Author):

The authors have addressed all my comments and I have no further concerns about the manuscript.